# CODED-SMOOTHING: CODING THEORY HELPS GENERALIZATION

## ABSTRACT

We introduce the coded-smoothing module, which can be seamlessly integrated into standard training pipelines, both supervised and unsupervised, to regularize learning and improve generalization with minimal computational overhead. In addition, it can be incorporated into the inference pipeline to randomize the model and enhance robustness against adversarial perturbations. The design of coded-smoothing is inspired by *general coded computing*, a paradigm originally developed to mitigate straggler and adversarial failures in distributed computing by processing linear combinations of the data rather than the raw inputs. Building on this principle, we adapt coded computing to machine learning by designing an efficient and effective regularization mechanism that encourages smoother representations and more generalizable solutions. Extensive experiments on both supervised and unsupervised tasks demonstrate that coded-smoothing consistently improves generalization and achieves state-of-the-art robustness against gradient-based adversarial attacks.

## 1 INTRODUCTION

Reliable prediction remains a central challenge in modern machine learning. Although deep neural networks have achieved remarkable success across computer vision, natural language processing, and reinforcement learning, their generalization beyond training data remains imperfect, and their reliability under adversarial perturbations is still limited (Szegedy et al., 2013; Goodfellow et al., 2014; Wen et al., 2020; Liu et al., 2020). This vulnerability is largely a consequence of overparameterization combined with limited training data, which makes models prone to overfitting, memorization, and brittle behavior when faced with unseen or corrupted inputs. Regularization techniques therefore play a key role in improving reliability: by guiding models toward simpler and smoother solutions, they reduce generalization error while simultaneously enhancing robustness to adversarial attacks.

Classical regularization strategies such as weight decay (Krogh & Hertz, 1991), dropout (Srivastava et al., 2014), and batch normalization (Ioffe & Szegedy, 2015) have long been established. More recently, data-centric approaches such as label smoothing (Szegedy et al., 2016), mixup and its variations (Zhang et al., 2017; Verma et al., 2019; Berthelot et al., 2019; Yun et al., 2019; Yao et al., 2022; Pinto et al., 2022; Bouniot et al., 2023) have become widely adopted for supervised learning. Nonetheless, data-centric approaches that are broadly applicable to both supervised and unsupervised models, and that simultaneously enhance generalization and adversarial robustness, remain insufficiently investigated.

In this paper, we take a step toward closing this gap, and introduce a new powerful regularization method, using *coded-smoothing module*, which applies seamlessly in both supervised and unsupervised settings. Our approach draws inspiration from an unexpected source: *coded computing*. Originally developed for distributed computing systems to mitigate the effects of straggler servers (Yu et al., 2017; 2020; Dutta et al., 2020; Jahani-Nezhad & Maddah-Ali, 2022; Moradi et al., 2024; Moradi & Maddah-Ali, 2025) and adversarial servers (Yu et al., 2019; Soleymani et al., 2022; Moradi et al., 2025), coded computing injects redundancy into the computational process. In this approach, instead of directly processing raw data and computing the designated results, the servers operate on carefully designed weighted linear combinations of the data, referred to as coded inputs. The number of coded inputs exceeds that of the original raw inputs. This coded redundancy enables the recovery of the original computation through a decoding procedure, even in the presence of missing results from stragglers or corrupted results from adversarial servers. In particular, in *general coded*

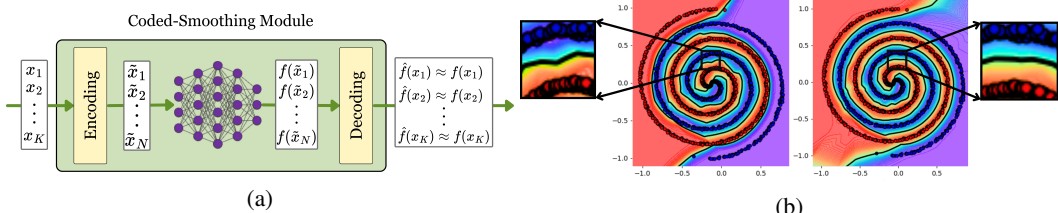

(a)

(b)

Figure 1: (a) In a coded computing module, instead of directly computing $f(x_1), \ldots, f(x_K)$, the system computes $f(\tilde{x}_1), \ldots, f(\tilde{x}_N)$, where $N > K$ and each coded input $\tilde{x}_i$ is a unique weighted linear combination of the originals. The desired outputs are then reconstructed via a decoding procedure, yielding approximations $\hat{f}(x_1) \approx f(x_1), \ldots, \hat{f}(x_K) \approx f(x_K)$. (b) Classification boundaries on the 2D spiral dataset, trained with Mixup (left) versus with the coded-smoothing module. The decision boundaries produced by the coded-smoothing model are noticeably smoother and less sensitive to individual data points, maintaining a more stable margin around the data.

*computing* (Moradi et al., 2024), the smoother the function representing the computation task, the more accurate the approximated result.

The coded-smoothing module has impactful structure. Given a batch of $K$ input samples, it first generates a new batch of $N$ *coded samples* through an encoding process, where each coded sample is formed as a combination of all inputs in the batch. The network is then evaluated on these coded samples, and a subsequent decoding step reconstructs estimates of the network outputs on the original inputs (see Figure 1a). Importantly, enforcing closeness between these decoded estimates and the true outputs induces local smoothness in the learned network and effectively reduces its complexity. To achieve this, during training we augment the objective with an auxiliary penalty term that encourages the decoded outputs to remain close to their true counterparts (see Fig. 2), thereby guiding the model toward smoother and more generalizable solutions (see Fig. 1b).

Beyond training, using coded-smoothing module offers a striking additional benefit at inference time. Since the coded-smoothing module works independently from the order of data in the input batch, we can inject randomness by applying a random shuffle before encoding and restoring the order after decoding. This simple yet powerful mechanism disrupts gradient-based adversarial attacks such as FGSM (Goodfellow et al., 2014) and PGD (Madry et al., 2017), which rely on precise gradient information to craft adversarial examples. As a result, the model attains substantially improved robustness against adversarial perturbations. Notably, this method imposes negligible computational overhead, making it both effective and practical for real-world deployment.

Our experiments show that the coded-smoothing module consistently improves generalization across a wide range of architectures and benchmarks in both supervised and unsupervised settings. Moreover, coded-smoothing provides substantial gains in adversarial robustness. Compared to mixup (Zhang et al., 2017), it achieves an 8.8% higher accuracy under the FGSM attack ($\epsilon = 8/255$) (Goodfellow et al., 2014), a 31.8% improvement under PGD with 10 steps, and a 37% improvement under PGD with 100 steps (Madry et al., 2017).

**Contributions.** In summary, this work makes the following key contributions:

- We introduce the *coded-smoothing* module, a novel and computationally efficient regularization mechanism for neural networks inspired by principles of coded computing (Section 3 and Appendix D).

- We provide a theoretical characterization showing that coded-smoothing enforces higher-order local smoothness, thereby acting as a powerful regularizer (Section 4.1).

- We propose a randomized coded inference procedure based on the coded-smoothing module that substantially improves adversarial robustness without requiring adversarial training (Section 5).

- We conduct extensive experiments demonstrating that coded-smoothing consistently enhances both generalization and robustness across datasets and architectures, while incurring minimal computational overhead (Section 6).

---

**Algorithm 1:** Pseudo-code for coded-smoothing module

---

**Input:** Input tensor $X$ of shape $(K, \cdot)$, where $K$ is the batch size; Computation function $f$ (e.g.,
      a neural network model).

**Output:** Estimated output tensor $\hat{f}(X)$ of shape $(K, \cdot)$.

```python
class CodedSmoothing(nn.Module):
    def __init__(self, K, N):
        super().__init__()
        self.alpha = generate_encoding_points(K)
        self.beta = generate_decoding_points(N)
        self.enc = Spline(knots=alpha)
        self.dec = Spline(knots=beta)
    def forward(self, X, f):
        self.enc.fit(self.alpha, X)
        x_coded = self.enc.predict(self.beta)
        f_coded = f(x_coded)
        self.dec.fit(self.beta, f_coded)
        f_hat = self.dec.predict(self.alpha)
        return f_hat
```

---

## 2 EMPIRICAL RISK MINIMIZATION (ERM)

In the supervised learning setting, let $\mathcal{D} := \{(x_i, y_i)\}_{i=1}^{n}$ denote a training dataset of size $n$, sampled from a distribution $\mathbb{P}$, where $x_i \in \mathcal{X}$ is the input and $y_i \in \mathcal{Y}$ is the corresponding label. Here, $\mathcal{X}$ and $\mathcal{Y}$ represent the input and output spaces, respectively, and $\theta \in \Theta$ denotes the parameter space. Given a loss function $\ell(\cdot, \cdot)$, ERM aims to learn a mapping $f_\theta : \mathcal{X} \to \mathcal{Y}$ by minimizing the expected loss with respect to the empirical distribution $\mathbb{P}_e(x, y) := \frac{1}{n} \sum_{i=1}^{n} \delta(x = x_i, y = y_i)$.

$$\theta^* = \arg\min_{\theta} \mathbb{E}_{\mathbb{P}_e(x,y)}[\ell(f_\theta(x), y)] = \int \ell(f_\theta(x), y) \; d\mathbb{P}_e(x, y) = \frac{1}{n} \sum_{i=1}^{n} \ell(f_\theta(x_i), y_i). \quad (1)$$

The goal is for the learned model to generalize well to unseen samples drawn from a test distribution $\mathbb{P}_t$, in both in-distribution ($\mathbb{P}_t = \mathbb{P}$) and out-of-distribution ($\mathbb{P}_t \neq \mathbb{P}$) settings.

## 3 CODED-SMOOTHING MODULE

Building on the general coded computing Moradi et al. (2024), we propose the *coded-smoothing* module as a regularization technique to model smoothness. We first describe the architecture of the proposed module, and then explain how coded-smoothing integrates into both the training and inference pipelines. This integration leads to improved generalization as well as enhanced adversarial robustness of the model.

### 3.1 ARCHITECTURE

The coded-smoothing module consists of three components: an encoder function $u_{\text{enc}} : \mathbb{R} \to \mathcal{U}$, a computation function $f : \mathcal{U} \to \mathcal{V}$, and a decoder function $u_{\text{dec}} : \mathbb{R} \to \mathcal{V}$. Here, $\mathcal{U}$ and $\mathcal{V}$ are the input and output domains of the function $f(\cdot)$, and $f$ may represent a machine learning model or a set of consecutive layers in a deep neural network. Given a batch of input data $\{x_1, \ldots, x_K\}$, the module produces an estimate of the computation function on these inputs, denoted by $\{\hat{f}(x_i)\}_{i=1}^{K}$.

The end-to-end process proceeds as follows:

(1) **Encoding:** the encoder function $u_{\text{enc}}$ is fitted to the set of points $\{(\alpha_i, x_i)\}_{i=1}^{K}$, where $\alpha_1 < \alpha_2 < \cdots < \alpha_K \in [-1, 1]$ are referred to as *encoding points*. Therefore,

$$u_{\text{enc}}(\alpha_i) = x_i, \quad , \forall i \in [K]. \quad (2)$$

Then, $N$ *coded samples* are generated by evaluating the encoder at another fixed set $\{\beta_j\}_{j=1}^N$ with $\beta_1 < \beta_2 < \cdots < \beta_N \in [-1, 1]$, called *decoding points* $\tilde{x}_j = u_{\text{enc}}(\beta_j)$, for $j \in [N]$. We note that each coded sample $\tilde{x}_j$ is a combination of the original input dataset $\{x_i\}_{i=1}^K$.

(2) **Computation:** In this step, $f(\tilde{x}_j)$, for $j = 1, \ldots, N$, are computed.

(3) **Decoding:** In this stage, first, decoder function $u_{\text{dec}}$ is fitted to the set of points $\{(\beta_j, f(\tilde{x}_j))\}_{j=1\mathcal{F}}^N$, therefore,

$$u_{\text{dec}}(\beta_j) = f(\tilde{x}_j) = f(u_{\text{enc}}(\beta_j)), \quad \forall j \in [N], \tag{3}$$

where the second equation follows from (2). If the decoder $u_{\text{dec}}(\cdot)$ generalizes well, then $u_{\text{dec}}(z) \approx f(u_{\text{enc}}(z))$, for all $z \in [-1, 1]$. In particular, at the encoding points, we have,

$$u_{\text{dec}}(\alpha_i) \approx f(u_{\text{enc}}(\alpha_i)) = f(x_i), \tag{4}$$

where the first approximation relies on the generalization ability of $u_{\text{dec}}$, and the second equation follows from (2). Thus, $u_{\text{dec}}(\alpha_i)$ approximates $f(x_i)$. We define $\hat{f}(x_i) \triangleq u_{\text{dec}}(\alpha_i)$, for $i \in [K]$.

Algorithm 1 presents PyTorch-style pseudo-code for the coded-smoothing module. As suggested by (Moradi et al., 2024), we use *natural cubic splines* (cubic smoothing splines with smoothing parameter of zero) for both the encoder and decoder.

With a careful choice of encoding and decoding points, the following lemma provides a bound on the approximation error of the coded-smoothing module.

**Lemma 1.** *For a coded-smoothingmodule with $N$ coded samples, we have:*

$$\frac{1}{K} \sum_{i=1}^K \left| \hat{f}(x_i) - f(x_i) \right|^2 \leq \frac{2C}{N^3} \left( \|u_{enc}'' \cdot f' \circ u_{enc}\|_{L^2(\Omega)}^2 + \|u_{enc}'' \cdot f' \circ u_{enc}\|_{L^2(\Omega)}^2 \right), \tag{5}$$

*for some constant $C$.*

For proof, see Appendix B. Lemma 1 highlights an important property of coded-smoothing module: The larger the number of coded samples $N$ or the smoother the function $f$, the smaller the mean squared estimation error.

**Spline representation.** Let $S_{\vec{t}, \vec{y}}(\cdot)$ denote the smoothing spline fitted on $\{(t_i, y_i)\}_{i=1}^n$, where $t_i \in \mathbb{R}$, $y_i \in \mathbb{R}^d$, $\vec{y} := [y_1, \ldots, y_n]^T$, and $\vec{t} := [t_1, \ldots, t_n]^T$. It is well-known that $S_{\vec{t}, \vec{y}}(z) = \sum_{i=1}^n y_i \phi(z, t_i)$, where $\phi(.,.)$ is the kernel of the second-order Sobolev space (i.e. functions with square-integrable derivatives up to order two). Thus, $S_{\vec{t}, \vec{y}}(\cdot)$ is a linear function of $\vec{y}$ (Wahba, 1975). Therefore, for any evaluation set $\vec{v} := [v_1, \ldots, v_m]^T$, there exists a matrix $A_{\vec{t}, \vec{v}} \in \mathbb{R}^{n \times m}$, which depends only on the knot set $\vec{t}$, the evaluation points $\vec{v}$, and the smoothing parameter $\lambda$ (but not on $\vec{y}$), such that $[S_{\vec{t}, \vec{y}}(v_1), \ldots, S_{\vec{t}, \vec{y}}(v_m)]^T = A_{\vec{t}, \vec{v}}^T \vec{y}$. Recall that in the coded-smoothing module, both the encoder and decoder are implemented using smoothing splines. Therefore, we have:

$$u_{\text{enc}}(z) = \sum_{i=1}^K x_i \phi(z, \alpha_i), \quad u_{\text{dec}}(z) = \sum_{j=1}^N f(\tilde{x}_j) \phi(z, \beta_j). \tag{6}$$

For detailed expression of matrix form of encoder and decoder functions, see Appendix C.

## 4 TRAINING REGULARIZATION USING THE CODED-SMOOTHING MODULE

We now describe how the coded-smoothing module can be integrated into the training pipeline of machine learning models to improve generalization. Since coded-smoothing does not require label information, it can be applied in both supervised (Section 6.1) and unsupervised (Section 6.2) settings.

Figure 2 illustrates the role of coded-smoothing during training. The computation function $f$ may represent the entire network or a part of the network, which we refer to as the *target block*. The

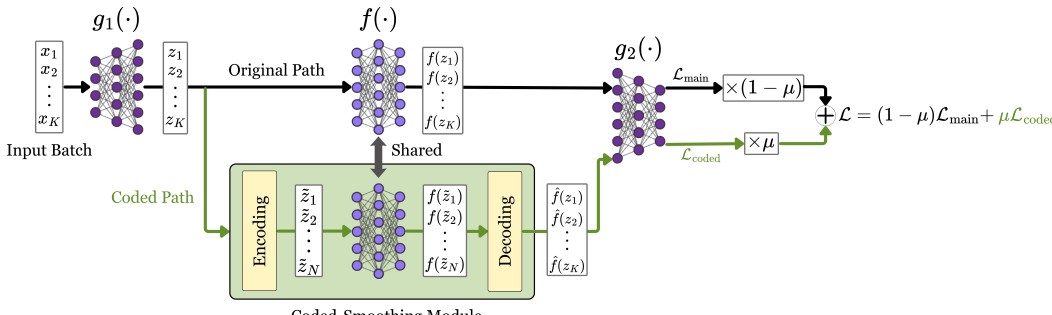

Figure 2: The proposed CODED-SMOOTHING as a regularization in training: the coded path includes a coded-smoothing module and runs in parallel to the original forward pass and contributes to the training objective.

integration of coded-smoothing introduces an additional *coded path* that runs in parallel to the original forward path.

Formally, consider training a deep neural network of the form $\text{net}(x) = g_1(f(g_2(x)))$, where $f(\cdot)$ is an intermediate target block. Suppose we apply coded-smoothing to $f(\cdot)$. After the input is passed through $g_1(\cdot)$, we branch it and follow two parallel paths: *the original path* and *the coded path* (see Fig. 2). In the coded path, there is a coded-smoothing module. The encoder generates a set of coded samples $\{\tilde{z}_j\}_{j=1}^N$, which form a new batch and are processed by the target block. The outputs of the target block on the coded samples are then passed through the decoder, producing estimated outputs $\{\hat{f}(z_i)\}_{i=1}^K$, which are approximately equal to $\{f(z_i)\}_{i=1}^K$. These estimated outputs are forwarded to the remainder of the network, denoted by $g_2(\cdot)$. During training, both paths contribute to the loss. Let $\mathcal{L}_{\text{main}}$ denote the loss from the original forward path, i.e., the standard training loss. Similarly, let $\mathcal{L}_{\text{coded}}$ denote the loss from the coded path, which has the same form as $\mathcal{L}_{\text{main}}$ but with the outputs of the original network replaced by those of the auxiliary coded path. The overall objective is then defined as

$$\mathcal{L} = (1 - \mu)\mathcal{L}_{\text{main}} + \mu\,\mathcal{L}_{\text{coded}}, \tag{7}$$

where $\mu \in [0, 1]$ is a weighting hyperparameter controlling the contribution of two paths. The parameters of the target block are shared between the original and coded paths, and the entire network is optimized with respect to the combined objective.

The second term in the loss function (7) acts as a regularizer, encouraging the coded path to match the predictive performance of the original path. In particular, it drives the coded-smoothing estimations of the target block toward their true outputs $\{f(z_i)\}_{i=1}^K$. Consequently, and in line with Lemma 1, the module implicitly enforces smoothness on the target block $f(\cdot)$. The effect of this regularization depends on the weighting coefficient $\mu$: when $\mu \approx 1$, training is dominated by the coded path, whereas when $\mu \approx 0$, the process reduces to training only with the original loss.

### 4.1 CODED-SMOOTHING IS A LOCAL HIGHER-ORDER SMOOTHER

In this subsection, we provide intuition for how the proposed approach encourages smoothness of the function. Recall from (4) that the accuracy of the approximation $\hat{f}(x_i) := u_{\text{dec}}(\alpha_i) \approx f(x_i)$ depends on the quality of the approximation $f(u_{\text{enc}}(z)) \approx u_{\text{dec}}(z)$. Moreover, from (6) we have $u_{\text{dec}}(z) = \sum_{j=1}^N f(\hat{x}_j)\,\phi(z, \beta_j)$. Hence, enhancing the approximation $\hat{f}(x_i) \approx f(x_i)$ is equivalent of improving the approximation

$$f(u_{\text{enc}}(z)) \approx \sum_{j \in [N]} f(\hat{x}_j)\,\phi(z, \beta_j). \tag{8}$$

The right-hand side is a weighted sum of some smooth functions, which implies that during training the regularized loss in (7) promotes smoothness of $f(u_{\text{enc}}(z))$, and consequently enforces smoothness in $f(\cdot)$ itself (see Fig. 3b).

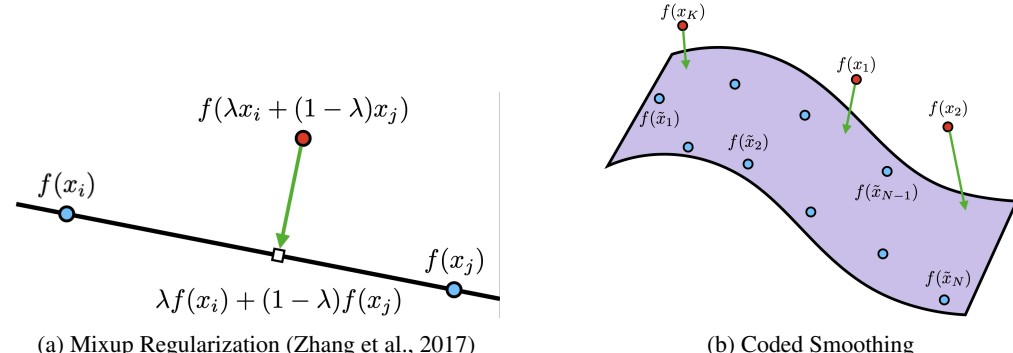

(a) Mixup Regularization (Zhang et al., 2017)  (b) Coded Smoothing

Figure 3: Coded Smoothing versus Mixup

To further clarify the concept, we next discuss the well-known Mixup method (Zhang et al., 2017) and highlight its connection to the proposed approach. In mixup, instead of empirical risk minimization, the model is trained by minimizing the expected loss with respect to a vicinal distribution $\mathbb{P}_v(x, y) := \frac{1}{n}\sum_{i=1}^n \delta(x = \bar{x}_i, y = \bar{y}_i)$, where $\bar{x}_i = \lambda x_i + (1 - \lambda)x_j$ and $\bar{y}_i = \lambda y_i + (1 - \lambda)y_j$ for $\lambda \sim$ Beta$(\alpha, \alpha)$. As a result, the model is encouraged to align the prediction $f(\lambda x_i + (1 - \lambda)x_j)$ with the target $\lambda y_i + (1 - \lambda)y_j$ for $\lambda \in [0, 1]$. At the endpoints ($\lambda = 0, 1$), this also recovers the original labels, i.e. $f(x_i) \approx y_i$ and $f(x_j) \approx y_j$. Consequently, training implicitly enforces local linearity on the model which regularizes $f$ to vary smoothly along the line segment connecting $f(x_i)$ and $f(x_j)$ (see Figure 3a):

$$f(\lambda x_i + (1 - \lambda)x_j) \approx \lambda f(x_i) + (1 - \lambda)f(x_j), \quad \lambda \in [0, 1]. \tag{9}$$

Comparing (8) and (9) reveals an intriguing connection between the two schemes. While the coded-smoothing module encourages $f(u_{\text{enc}}(z))$ to approximate a linear combination of smooth functions, mixup explicitly encourages $f(\cdot)$ to behave like a linear function. In other word, coded-smoothing module imposes a higher-order smoothness constraint on $f$, regularizing it beyond pairwise linearity. Although both approaches promote smoothness in $f(\cdot)$, coded smoothness admits a richer structure and may potentially lead to improved generalization (See Section 6 on experiment results).

## 5 ROBUST INFERENCE USING A RANDOMIZED CODED-SMOOTHING MODULE

After training a model with the coded-smoothing module, both the coded path and the original path can be used during inference. Since the coded path generates a smooth approximation of the original outputs, its standalone generalization performance is dominated by that of the original path. However, the coded path possesses a useful property that can be exploited to substantially enhance adversarial robustness.

The key observation is that the proposed module performance does not depend on the order of input samples within a batch: the coded-smoothing module generates a good estimate for each input regardless of its position in the batch. During training, due to random shuffling across epochs, each sample $x_i$ appears at different indices and the network aligns the estimation $\hat{f}(x_i)$ with its true output $f(x_i)$ independently of the sample's index.

Consequently, at inference time, one can introduce additional randomness by applying a random permutation $\pi$ to the batch before feeding it into the encoder, and subsequently restoring the original order using $\pi^{-1}$ before passing the outputs to the remainder of the network. We refer to this approach as *Randomized Coded Inference (RCI)*. Figure 4 illustrates this inference approach.

This strategy disrupts adversarial attacks, particularly gradient-based methods such as FGSM (Goodfellow et al., 2014) and PGD (Madry et al., 2017), which rely on precise gradients to craft adversarial examples. The core idea in these methods is to generate an adversarial sample by perturbing the input in the direction of the gradient of the loss with respect to that input. However, since $\pi$ is chosen uniformly at random from all permutations, with high probability the permutation used by the network at inference differs from the one assumed by the adversary when generating the perturbations.

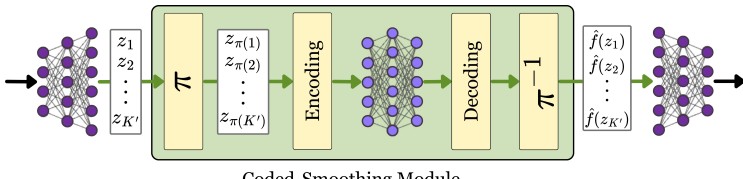

Figure 4: The proposed RANDOMIZED CODED INFERENCE: $\pi$ represents a random permutation.

As a result, the network's robustness is significantly improved. Note that although coded-smoothing operates in batch mode at inference, the batch size need not match that used during training. In practice, the method is effective for batch sizes as small as $K' \geq 4$, since spline fitting requires at least three points, thereby offering flexibility for deployment (see Table 10 in Appendix H).

## 6 EXPERIMENTS

In this section, we evaluate the performance of the proposed coded-smoothing training method (using the coded-smoothing module) as well as the randomized coded inference approach, under various settings and across multiple evaluation metrics. We begin with the supervised scenario (Section 6.1), followed by the unsupervised setting (Section 6.2). We then demonstrate how coded-smoothing substantially enhances adversarial robustness during inference (Section 6.3). Finally, we assess its effectiveness under distribution shift, where the test distribution differs from the training distribution (Section 6.4). All experiments are conducted in PyTorch (Paszke et al., 2019) on a single machine equipped with an NVIDIA RTX 5090 GPU.

In all experiments, following Jahani-Nezhad & Maddah-Ali (2022); Moradi et al. (2024), we adopt the first-order Chebyshev points for encoding and the second-order Chebyshev points for decoding, i.e., $\alpha_i = \cos(\frac{(2i-1)\pi}{2K})$ and $\beta_j = \cos(\frac{(j-1)\pi}{N-1})$ for $i \in [K]$, $j \in [N]$. This choice is motivated by their superior empirical performance (Jahani-Nezhad & Maddah-Ali, 2022) and desirable theoretical properties in approximation theory (Phillips, 2003; Trefethen, 2019).

### 6.1 SUPERVISED

We begin by evaluating the effectiveness of the coded-smoothing module in the supervised learning setting. In particular, we compare its generalization performance against standard empirical risk minimization (ERM) as well as two widely used mixup-based regularization methods, mixup (Zhang et al., 2017) and manifold mixup (Verma et al., 2019).

**Datasets and architectures.** To ensure a comprehensive evaluation across model families and dataset complexities, we conduct experiments on CIFAR-10, CIFAR-100 (Krizhevsky et al., 2009), TinyImageNet (Le & Yang, 2015), and Imagenet-1k (Russakovsky et al., 2015). For CIFAR-10, we use PreActResNet18 (He et al., 2016); for CIFAR-100, we employ WideResNet28-10 (Zagoruyko & Komodakis, 2016); and for TinyImageNet and Imagenet, we adopt ResNet50 (Goyal et al., 2017). These architectures are chosen to capture a range of model complexities while aligning with prior work.

In all supervised experiments, we empirically find that the best performance is achieved when the coded-smoothing module is applied to the full network. Table 1 reports the test performance across datasets and architectures. Each experiment is repeated over 5 independent train-validation splits with different random seeds. We report both the mean and standard deviation. As shown in Table 1, training with coded-smoothing consistently outperforms both mixup and ERM baselines across all benchmarks. Additional experiments and hyperparameter selection are provided in Appendix E.1.

### 6.2 UNSUPERVISED

Next, we take one step further and evaluate the effectiveness of the CODED-SMOOTHING training in an unsupervised setting. Specifically, we incorporate coded-smoothing into the training of a WGAN-GP(Gulrajani et al., 2017) which is a variant of WGAN (Arjovsky et al., 2017).

Table 1: Comparisons of accuracies (%) on in-distribution test data.

| | CIFAR-10 PARN18 | CIFAR-100 WRN28-10 | TinyImageNet RN50 | ImageNet RN50 |
|---|---|---|---|---|
| ERM | $93.8 \pm 0.2$ | $76.7 \pm 0.3$ | $62.9 \pm 0.9$ | $69.5$ |
| Mixup | $95.6 \pm 0.2$ | $80.2 \pm 0.3$ | $65.4 \pm 1.0$ | $69.1$ |
| Manifold Mixup | $95.43 \pm 0.12$ | $\mathbf{81.1 \pm 0.4}$ | $\mathbf{67.4 \pm 0.3}$ | $67.6$ |
| CODED-SMOOTHING (ours) | $\mathbf{95.8 \pm 0.1}$ | $79.9 \pm 0.4$ | $\mathbf{67.1 \pm 0.5}$ | $\mathbf{70.1}$ |

Table 2: Comparison of FID and IS for generated images for CIFAR-10 and CelebA.

| Method | CIFAR-10 | | CelebA |
|---|---|---|---|
| | IS | FID | FID |
| WGAN-GP | $7.08 \pm 0.07$ | $26.93 \pm 0.61$ | $28.22 \pm 0.17$ |
| WGAN-GP + CODED-SMOOTHING | $\mathbf{7.38 \pm 0.06}$ | $26.94 \pm 0.89$ | $\mathbf{24.58 \pm 0.62}$ |

Prior work has shown that regularizing the discriminator can improve GAN training stability and performance (Zhang et al., 2017; Verma et al., 2019). However, because mixup and its variants rely on label information, they cannot be directly applied to the generator. Here we use CODED-SMOOTHING training method to regularize the *generator* of a WGAN. Specifically, we use coded-smoothing module with $N = K$ with batchsize $K = 64$ and $\mu = 0.5$. Further experimental details are provided in Appendix E.2. Table 2 reports the Fréchet Inception Distance (FID) (Heusel et al., 2017) and Inception Score (IS) (Salimans et al., 2016) on the CIFAR-10 and CelebA (Liu et al., 2018) datasets, which serve as standard metrics for evaluating generative quality and generalization. As shown in the results, regularizing generator with improves FID and IS, indicating enhanced generalization and higher-quality image generation.

## 6.3 ADVERSARIAL ROBUSTNESS

We next evaluate the effectiveness of randomized coded inference (RCI) against adversarial attacks on CIFAR-10, focusing on FGSM (Goodfellow et al., 2014) and PGD (Madry et al., 2017) attacks. Since the coded-smoothing module is non-parametric, RCI can be applied to the inference stage of any trained model, with the number of coded samples $N$ adjusted independently of training. Importantly, $N$ can be set relative to the batch size without incurring significant performance degradation (see Table 10 in Appendix H.5 for a sensitivity analysis with respect to batch size).

As shown in Table 3, RCI substantially improves adversarial robustness across all methods, including models already trained with CODED-SMOOTHING, while incurring only a marginal drop in clean (no-attack) accuracy. The strongest results are achieved when models are trained with CODED-SMOOTHING and evaluated with RCI using $N = 1.5K$, where $K = 128$ is the batch size. In this setting, the generalization error increases by only $1\%$, but robustness gains are significant: improvements of $+8.8\%$ under FGSM ($\epsilon = 8/255$), $+33\%$ under PGD with 10 steps, and $+5.4\%$ under PGD with 100 steps compared to mixup. These results highlight the effectiveness of using RCI in inference for adversarial robustness.

## 6.4 COVARIATE SHIFT ROBUSTNESS

Finally, we evaluate the performance of the proposed method under distribution shift, where the test distribution differs from the training distribution. For this evaluation, we use CIFAR-10.1 (Recht et al., 2018) and CIFAR-10.2 (Lu et al., 2020), which represent natural covariate shifts of CIFAR-10, CIFAR-10C (Hendrycks & Dietterich, 2019), which introduces 19 types of synthetic corruptions applied at 5 levels of severity to the CIFAR-10 test set, and ImageNet-R (Hendrycks et al., 2021), which contains multiple renditions of ImageNet classes. Table 5 in Appendix F compares the performance of our method against ERM and mixup. The coded-smoothing module consistently outperforms both baselines on CIFAR-10.1, CIFAR-10.2, and ImageNet-R, and achieves comparable performance on CIFAR-10C. For CIFAR-10C, accuracy is reported as the average of all corruption.

Table 3: Comparison of CIFAR-10 test accuracies under adversarial attacks, contrasting randomized coded inference (RCI) with standard inference. Manifold mixup results are reported from (Verma et al., 2019).

| Inference method | Training Method | No Attack - | FGSM $\epsilon = \frac{8}{255}$ | PGD 10 steps | PGD 100 steps |
|---|---|---|---|---|---|
| Standard inference | ERM | 93.7 | 36.5 | 5.5 | 0.0 |
| | Mixup | 95.5 | 71.7 | 39.9 | 0.4 |
| | Manifold Mixup | 95.2 | 61.7 | 30.9 | 0.0 |
| | CODED-SMOOTHING (ours) | **95.8** | 47.7 | 8.6 | 0.0 |
| RCI ($N = 128$) | ERM | 55.3 | 49.1 | 46.8 | 19.4 |
| | Mixup | 72.4 | 66.1 | 64.1 | **37.4** |
| | CODED-SMOOTHING(ours) | 72.4 | 66.2 | 63.5 | 27.7 |
| RCI ($N = 190$) | ERM | 90.2 | 75.8 | 65.7 | 6.3 |
| | Mixup | 93.5 | 78.2 | 65.1 | 9.9 |
| | CODED-SMOOTHING(ours) | 94.8 | **80.5** | **72.0** | 5.8 |

## 7 RELATED WORK

Improving generalization has long been a central challenge in machine learning research. A first class of methods enhances generalization by perturbing hidden representations during training. Classical examples include dropout (Srivastava et al., 2014) and batch normalization (Ioffe & Szegedy, 2015), both of which reduce overfitting by encouraging more robust internal representations.

A second major line of research focuses on data augmentation. Among these, mixup (Zhang et al., 2017) has become a widely adopted regularization strategy. Since its introduction, numerous variants have been proposed to address different limitations of mixup, such as improving generalization (Verma et al., 2019; Yun et al., 2019), adapting it to regression tasks (Yao et al., 2022), enhancing robustness to distribution shift (Pinto et al., 2022), and improving calibration (Bouniot et al., 2023). Despite these extensions, all mixup-style methods fundamentally rely on label information and are thus not applicable in unsupervised settings. The only exception is in GANs (Goodfellow et al., 2020), where mixup regularization has been applied to the supervised discriminator module (Zhang et al., 2017; Verma et al., 2019).

To partially address this limitation, Verma et al. (2022) proposed an unsupervised mixup loss for semi-supervised problems. Their method encourages local linearity by explicitly enforcing the mixup interpolation constraint (see Figure 3). While effective, this approach enforces only pairwise linear constraints, limiting its ability to capture higher-order structures.

In contrast, the proposed coded-smoothingmodule provides a unified regularization framework applicable to both supervised and unsupervised settings with negligible computational overhead. Beyond enforcing linearity, it imposes higher-order smoothness. Moreover, through randomized coded inference, coded-smoothingachieves state-of-the-art robustness against adversarial attacks.

## 8 CONCLUSION

In this paper, we introduced the coded-smoothing module, a novel regularization framework inspired by coded computing. By enforcing local higher-order smoothness during training, coded-smoothing promotes more generalizable and reliable models. At inference, random shuffling within coded-smoothing, randomized coded inference (RSI), significantly enhances adversarial robustness.

Our method is computationally efficient and applicable to both supervised and unsupervised learning. Across benchmarks and architectures, coded-smoothing improves supervised generalization, outperforming ERM and mixup, while achieving state-of-the-art robustness to adversarial attacks with minimal overhead. In unsupervised settings, applying coded-smoothing to GAN generators boosts generative quality, demonstrating its effectiveness as a label-free regularizer.

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

# A APPENDIX

## A.1 CODED COMPUTING

Inspired by the success of coding theory in communication over unreliable channels, *coded computing* has emerged as an efficient framework for distributed computation. It addresses key challenges in distributed systems, particularly the presence of straggling or adversarial servers (Yu et al., 2017). Early work introduced coded computing for fundamental tasks such as polynomial evaluation (Yu et al., 2019; Fahim & Cadambe, 2021) and matrix multiplication (Yu et al., 2017; Jahani-Nezhad & Maddah-Ali, 2021). More recently, Moradi et al. (2024; 2025); Moradi & Maddah-Ali (2025) proposed a framework grounded in learning theory that extends coded computing to a wide range of functions, including deep neural networks, with provable resilience against stragglers and adversaries.

The central idea of coded computing is to assign each server a (linear) combination of the data, referred to as coded data, instead of the raw inputs. The number of coded symbols exceeds that of the original data, effectively over-representing the data. This redundancy is then leveraged to mitigate the effects of stragglers and adversarial behavior. Formally, suppose a master node aims to approximately compute a function $f : \mathcal{X} \to \mathcal{Y}$ on a dataset $\{x_1, \dots, x_K\}$ using a cluster of $N$ servers, some of which may be stragglers. The coded computing framework proceeds in three steps:

(1) **Encoding:** The master node fits an encoder function $u_{\text{enc}} : \mathbb{R} \to \mathcal{X}$ to the set of points $\{(\alpha_i, x_i)\}_{i=1}^K$, where $\alpha_1 < \alpha_2 < \cdots < \alpha_K \in \Omega \subset \mathbb{R}$ are referred to as *encoding points*. Thus,

$$\forall i \in [K], \quad u_{\text{enc}}(\alpha_i) \approx x_i. \tag{10}$$

The master node then generates $N$ *coded data* by evaluating the encoder at another fixed set $\{\beta_j\}_{j=1}^N$ with $\beta_1 < \beta_2 < \cdots < \beta_N \in \Omega \subset \mathbb{R}$, called *decoding points*:

$$\tilde{x}_j = u_{\text{enc}}(\beta_j), \quad j \in [N]. \tag{11}$$

Each coded point $\tilde{x}_j$ is a combination of the original input dataset $\{x_i\}_{i=1}^K$, and is then assigned to server $j$.

(2) **Computation:** Each server $j$ computes $f(\tilde{x}_j)$ and returns the result to the master. Due to the presence of stragglers, some results may be missing. Let $\mathcal{F}$ denote the set of indices corresponding to successfully returned results.

(3) **Decoding:** Given the received outputs $\{f(\tilde{x}_v)\}_{v \in \mathcal{F}}$ from the non-straggler servers, the master node fits a decoder function $u_{\text{dec}} : \mathbb{R} \to \mathcal{Y}$ at the points $\{(\beta_v, f(\tilde{x}_v))\}_{v \in \mathcal{F}}$. Consequently,

$$\forall j \in [N], \quad u_{\text{dec}}(\beta_j) \approx f(u_{\text{enc}}(\beta_j)). \tag{12}$$

If the decoder $u_{\text{dec}}(\cdot)$ generalizes well, it can approximate $f(\cdot)$ on the original dataset $\{x_i\}_{i=1}^K$:

$$\hat{f}(x_i) \triangleq u_{\text{dec}}(\alpha_i) \approx f(u_{\text{enc}}(\alpha_i)) \approx f(x_i), \tag{13}$$

where the first approximation relies on the generalization ability of $u_{\text{dec}}$, and the second follows from (10).

The goal is to obtain an accurate estimate of the function $f(\cdot)$ on the input dataset. The key design choice in the coded computing scheme is selecting encoder and decoder functions that yield low estimation error.

Moradi et al. (2024) propose using smoothing splines (Wahba, 1975; 1990) as both encoder and decoder functions, fitted on $\{x_i\}_{i=1}^K$ and $\{f(\tilde{x}_v)\}_{v \in \mathcal{F}}$ with smoothing parameters $\lambda_e$ and $\lambda_d$, respectively. More specifically, the decoder is obtained by solving the following optimization problem:

$$u_{\text{dec}}^\star = \arg\min_{u \in \mathcal{H}^2(\Omega)} \frac{1}{|\mathcal{F}|} \sum_{v \in \mathcal{F}} \|u(\beta_v) - f(u_{\text{enc}}(\beta_v))\|^2 + \lambda_d \cdot \|u''\|_{L^2(\Omega)}^2, \tag{14}$$

where $\mathcal{H}^2(\Omega)$ denotes the second-order Sobolev space over $\Omega$ (i.e., functions with square-integrable derivatives up to order two), and $\|\cdot\|_{L^2(\Omega)}$ denotes the $L^2$-norm on $\Omega$.

Using the decoder function in (14), together with a careful choice of encoding points, decoding points, and smoothing parameter, Moradi et al. (2024) show that the mean squared estimation error of the coded computing scheme can be upper bounded as follows:

**Theorem 1** (Moradi et al. (2024)). *Consider the coded computing framework with $N$ servers and at most $S$ stragglers. Suppose $\Omega = (-1, 1)$ and $f(\cdot)$ is $\mu$-Lipschitz continuous. Then,*

$$\frac{1}{K} \sum_{i=1}^{K} \left\| \hat{f}(x_i) - f(x_i) \right\|^2 \leq C \left( \frac{S+1}{N} \right)^3 \|(f \circ u_{enc})''\|_{L^2(\Omega)}^2 + \frac{2\mu^2}{K} \sum_{k=1}^{K} \|u_{enc}(\alpha_k) - x_k\|^2, \tag{15}$$

*where $C > 0$ is a constant.*

## B    PROOF OF LEMMA 1

Since in the coded-smoothingmodule $u_{\text{enc}}(\alpha_i) = x_i$, the second term in Theorem 1 vanishes. Setting $S = 0$ and applying the chain rule to $(f \circ u_{\text{enc}})''$ completes the proof of the lemma.

## C    ENCODER AND DECODER MATRIX REPRESENTATION

Here also, we can present a matrix representation of the encoding and decoding processes. We define $A^{\text{enc}} := A_{\vec{\alpha}, \vec{\beta}} \in \mathbb{R}^{K \times N}$ and $A^{\text{dec}} := A_{\vec{\beta}, \vec{\alpha}} \in \mathbb{R}^{N \times K}$ as the encoder and decoder matrices, respectively, where $\vec{\alpha} := [\alpha_1, \ldots, \alpha_K]^T$ and $\vec{\beta} := [\beta_1, \ldots, \beta_N]^T$ denote the encoding and decoding points, respectively. The coded samples are then obtained as

$$\forall j \in [N], \quad \tilde{x}_j = \langle A_j^{\text{enc}}, \vec{x} \rangle \tag{16}$$

where $A_j^{\text{enc}}$ denotes the $j$th column of the matrix $A^{\text{enc}}$ and $\vec{x} = [x_1, \ldots, x_K]^T$. Finally, letting $\vec{f}(A^{\text{enc}}, \vec{x}) := [f(\langle A_1^{\text{enc}}, \vec{x} \rangle), \ldots, f(\langle A_N^{\text{enc}}, \vec{x} \rangle)]^T$, the decoded estimates are given by

$$\forall i \in [K], \quad \hat{f}(x_i) = \langle A_i^{\text{dec}}, \vec{f}(A^{\text{enc}}, \vec{x}) \rangle. \tag{17}$$

## D    COMPUTATIONAL COMPLEXITY OF CODED-SMOOTHING MODULE

Since both the encoder and decoder functions are non-parametric, the coded-smoothing module does not introduce any additional learnable parameters to the model.

Additionally, when $N = K$, the coded path has approximately the same computational cost as the original path. The only extra overhead comes from the encoding and decoding operations (i.e., evaluating and fitting the splines), which contribute only a negligible cost compared to the main computation. This efficiency arises because fitting and evaluating smoothing splines can be performed in linear time using the B-spline basis representation (De Boor, 2001; Eilers & Marx, 1996). More specifically, if the input and output dimensions are $d$ and $m$ (for example, when applying coded smoothing to the entire network on CIFAR-10, the input dimension is $32 \times 32$ and the output dimension is 10), the computational complexities of the encoding and decoding steps are $\mathcal{O}((N + K) \cdot d)$ and $\mathcal{O}((N + K) \cdot m)$, respectively. Assuming $N \approx K$, both terms scale linearly in the dimension and batch size and are negligible relative to the main computation.

To illustrate this in practice, we conduct a runtime analysis to compare the computational cost of the coded path and the original path. Table 4 reports the runtime for processing a batch of 128 CIFAR-10 images using PreActResNet18 on a single NVIDIA RTX 5090 GPU.

These results show that the coded-smoothing module introduces only minimal computational overhead in practice.

Table 4: Inference time comparison between the original and coded paths.

| Method | Time |
|---|---|
| Original Path | $1.5\,\text{ms} \pm 0.5$ |
| Coded Path | $2\,\text{ms} \pm 0.4$ |

## E EXPERIMENTAL DETAILS

### E.1 SUPERVISED

For all supervised learning experiments, we train models on the CIFAR-10, CIFAR-100, Tiny ImageNet, and ImageNet-1k datasets under consistent optimization settings.

In the CIFAR-10 and CIFAR-100 experiments, we train for 350 epochs with an initial learning rate of $0.1$. The learning rate is decayed by a factor of 10 at epochs 100 and 250 for both datasets.

For Tiny ImageNet, we also use an initial learning rate of $0.1$. The learning rate is decayed by a factor of 10 at epochs 100, 200, and 300.

For ImageNet-1k, we follow the training schedule reported in Zhang et al. (2017). We use an initial learning rate of $0.2$, and decay it by a factor of 10 at epochs 30, 60, and 90.

In all experiments, we use stochastic gradient descent (SGD) with momentum 0.9 as the optimizer, and set the batch size to 128.

#### E.1.1 HYPERPARAMETER SELECTION

For mixup and manifold mixup, the mixing coefficient $\lambda$ is sampled from a Beta distribution $\text{Beta}(\alpha, \alpha)$, where $\alpha$ is chosen according to the best-performing settings reported in prior work (Zhang et al., 2017; Verma et al., 2019). Specifically, for mixup we use $\alpha = 1.0$ on CIFAR-10 and CIFAR-100, and $\alpha = 0.2$ on TinyImageNet and ImageNet-1k. For manifold mixup, we select the set of intermediate layers following the best-performing configuration reported in Verma et al. (2019), and set $\alpha = 2.0$ for CIFAR-10 and CIFAR-100, and $\alpha = 0.2$ for TinyImageNet and ImageNet-1k.

Training with the CODED-SMOOTHING method introduces two hyperparameters: $\mu$ and $N$. The parameter $\mu$ in (7) balances the contribution of the coded path in the overall loss, while $N$ controls the accuracy of the estimation. From Lemma (1), the discrepancy between the original and coded path outputs decreases either as $N$ increases or as $f(\cdot)$ becomes smoother. However, setting $N$ too large makes the coded and original paths nearly identical, effectively collapsing the method to ERM. Empirically, we find that initializing with $N = K$ (the batch size, see Figure 2) and $\mu = 0.5$ yields the best trade-off between regularization strength and predictive accuracy. A detailed sensitivity analysis of $N$ and $\mu$ is presented in Tables 6 and 7 in Appendix H.

**Scheduling $N$.** Fixing $N$ during training causes saturation, as the gap between $f$ and $\hat{f}$ stops shrinking once a certain smoothness is reached. To address this, we gradually increase $N$ from the batch size $K$ to $\gamma K$ with $\gamma > 1$. We find $\gamma = 1.5$ works best, yielding two benefits: improved coded-path accuracy (making it reliable for inference) and escaping training plateaus for further gains (see Figure 5a).

### E.2 UNSUPERVISED

For the unsupervised experiments with WGAN, we used the following configuration: the training was performed for 100,000 iterations with a batch size of 64. The number of coded points ($N$) was set to 96. Moreover $\mu = 0.5$. The initial learning rate was $2 \times 10^{-4}$, and the critic was updated 5 times per generator step. We employed a gradient penalty coefficient $\lambda_{gp} = 10$, and the optimizer was Adam with betas $(0.0, 0.9)$. For Inception Score (IS) computation, 50,000 samples were used. For FID computation, we followed the standard protocol by comparing the statistics of 50,000 generated images with the real dataset.

## F    COVARIATE SHIFT ROBUSTNESS

Table 5: Comparisons of accuracies (%) on out-of-distribution test data.

|  | CIFAR-10.1 | CIFAR-10.2 | CIFAR-10.C | ImageNet-R |
|---|---|---|---|---|
| ERM | $86.5 \pm 0.4$ | $82.8 \pm 0.1$ | $72.7 \pm 0.3$ | 21.3 |
| Mixup | $88.9 \pm 0.4$ | $85.7 \pm 0.2$ | $\mathbf{78.2 \pm 0.3}$ | 19.8 |
| CODED-SMOOTHING (ours) | $\mathbf{89.6 \pm 0.5}$ | $\mathbf{86.4 \pm 0.1}$ | $77.6 \pm 0.2$ | $\mathbf{22.8}$ |

## G    PERFORMANCE DURING TRAINING

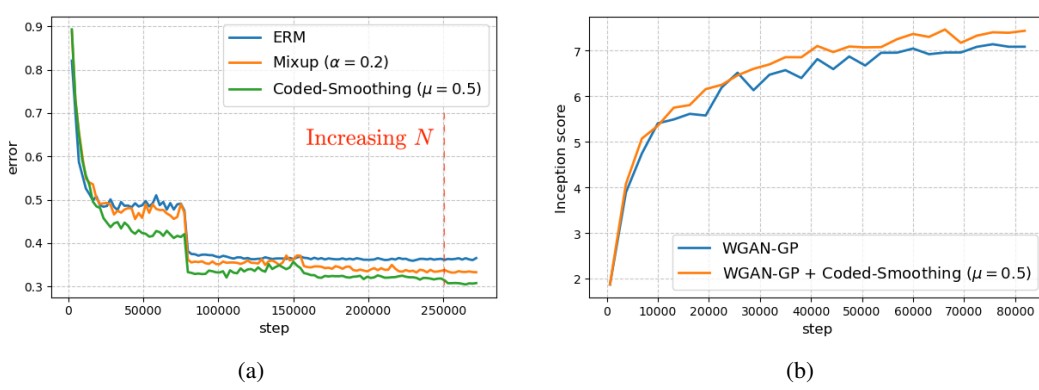

(a)                                                              (b)

Figure 5: (a) TinyImageNet validation loss for different methods during training. (b) Comparison of Inception Score (IS) during training on the CIFAR-10 dataset. In WGAN-GP + CODED-SMOOTHING, the coded-smoothing module is applied to the generator of the GAN architecture.

## H    ABLATION STUDY

### H.1    EFFECT OF NUMBER OF CODED SAMPLES ($N$)

Table 6: Test accuracy (%) of training with coded-smoothing module for different number of coded samples ($N$) on CIFAR-10 with batch size 128.

| $N$ | Acc |
|---|---|
| 110 | 95.6 |
| 130 | 95.0 |
| 150 | 94.8 |
| 170 | 94.5 |
| 190 | 94.2 |

### H.2    EFFECT OF $\mu$

Table 7: Test accuracy (%) of training with coded-smoothing module for different $\mu$ on CIFAR-10.

| $\mu$ | Acc |
|---|---|
| 0.1 | 95.1 |
| 0.2 | 95.4 |
| 0.4 | 95.7 |
| 0.5 | 95.9 |
| 0.6 | 95.8 |
| 0.8 | 95.4 |
| 1.0 | 95.0 |

## H.3 Effect of the layer set

We conducted additional experiments in which coded-smoothing was selectively applied to different layers or blocks within the ResNet architecture. Specifically, we applied the coded-smoothing module to various subsets of blocks in the PreActResNet-18 model for the CIFAR-10 task. The results are shown in Table 8.

Table 8: Effect of coded-path block selection on test accuracy and loss.

| Set of blocks in the coded path | Test Acc | Test Loss |
|:---:|:---:|:---:|
| 0 | 94.6 | 0.25 |
| 3 | 94.6 | 0.26 |
| 0,1 | 94.8 | 0.23 |
| 0,2 | 94.8 | 0.23 |
| 1,3 | 94.8 | 0.235 |
| 0,1,2 | 94.5 | 0.24 |
| 0,1,2,3 | 95.1 | 0.19 |
| 0,1,2,3,4 | 95.9 | 0.19 |

## H.4 Integrating with dropout

We evaluate the coded-smoothing module both with and without dropout. Following the recommendation in (Kim et al., 2023), for best performance, dropout should be inserted after the second batch-normalization layer in each ResNet block. The results demonstrate that dropout integrates smoothly with the proposed module and yields a modest improvement in generalization performance.

Table 9: Effect of integrating dropout on test accuracy and loss for the CIFAR-10 dataset using ERM and coded-smoothing.

| Method (CIFAR-10, PARN18) | Test Acc | Test Loss |
|:---|:---:|:---:|
| Raw | 93.780 | 0.308 |
| Raw + Dropout | 93.840 | 0.325 |
| Coded Smoothing | 95.120 | 0.240 |
| Coded Smoothing + Dropout | **95.200** | **0.217** |

## H.5 Effect of batch size and number of coded samples in RCI

Table 10: Comparison of accuracies of inference with coded path for the mode methods under Adversarial attack on CIFAR-10 dataset.

| Batch Size | # Coded Samples ($N$) | No Attack | FGSM ($\epsilon = \frac{8}{255}$) | PGD (10 steps) |
|:---:|:---:|:---:|:---:|:---:|
| 8 | 12 | 94.9 | 78.1 | 69.8 |
| 16 | 24 | 94.8 | 79.0 | 70.3 |
| 32 | 48 | 94.9 | 79.3 | 70.6 |
| 64 | 96 | 94.9 | 80.1 | 71.5 |
| 128 | 190 | 94.9 | 80.5 | 72.0 |

