# OpenReview forum: "Coded-Smoothing Module: Coding Theory Helps Generalization"
_ICLR.cc/2026/Conference — Submitted to ICLR 2026_

### Official Review · Reviewer_fg2C · 2025-10-27

**Soundness:** 3
**Presentation:** 3
**Contribution:** 3
**Rating:** 4
**Confidence:** 3

**Summary:**

This paper presents a novel training method for improving model generalization, drawing on the concept of coded computing. The approach involves recoding the input data and using it as an auxiliary branch for regularization during training, thereby enhancing the model's generalization capability.

**Strengths:**

1. The proposed regularization method is both novel and supported by a comprehensive theoretical analysis.
2.The proposed Coded Smoothing module is designed for flexible integration into both supervised and unsupervised training frameworks, and can be readily reproduced due to its straightforward implementation.

**Weaknesses:**

1. The method description in this paper is thorough, but the experiment appears relatively weak. The classification experiments should be supplemented with results on real-world datasets. Even under computational constraints, validation on a subset such as ImageNet-100[1] should be considered as a minimum requirement.
2. The generalization performance of the proposed method on OOD data remains insufficiently validated. It would be more convincing to include additional evaluations on specialized OOD benchmarks such as ImageNet-R and ImageNet-S.
3. The comparison with baseline methods remains relatively limited, as only classical approaches including ERM and Mixup are included. It would be beneficial to incorporate more recent and advanced baselines to better demonstrate the superiority of the proposed method.
4. The experimental evaluation of the generative model is currently insufficient, as quantitative metrics alone cannot fully capture the perceptual quality of generated samples. It is essential to include qualitative visualizations of the generated outputs. Furthermore, the analysis should at minimum demonstrate the improvement achieved over Mixup when applied in conjunction with WGAN.
5. The data encoding and decoding process introduces dimensionality expansion to the input. Has there been any analysis on the computational efficiency of this approach, particularly regarding the regularization overhead during training and the additional operations required during inference?

[1] Tian Y, Krishnan D, Isola P. Contrastive multiview coding[C]//European conference on computer vision. Cham: Springer International Publishing, 2020: 776-794.

**Questions:**

I will consider adjusting my score based on the authors' response to these weakness points.

---

> ### Author Response · Authors · 2025-11-21
>
> We sincerely thank the reviewer for their thoughtful and encouraging feedback.
>
> **Weaknesses:**
>
> >The method description in this paper is thorough, but the experiment appears relatively weak ...
>
> To further strengthen the empirical evaluation, we additionally trained **ResNet-50 on ImageNet-1K** following the Mixup paper’s recommended settings (α = 0.2; 90 epochs; initial LR = 0.2 with decays at epochs 30, 60, 80). The results are:
>
> |Method|Test Loss|Test Acc|
> |-|-|-|
> | Coded Smoothing |**1.20**|**70.1**|
> | Mixup|1.26|69.1|
> | Raw| 1.25|69.5|
>
> We will include these ImageNet-1K results in the revised version.
>
>
> >The generalization performance of the proposed method on OOD ...
>
> As reviewer suggested, we extend our distribution shift evaluation on Imagenet-R dataset. We use the trained model on imagenet-1k dataset to see their performance on Imagenet-R dataset. The results is as follows:
>
> |Method|ImagenNet-R Acc|
> |-|-|
> | Coded Smoothing |**22.8**|
> | Mixup|19.8|
> | Raw| 21.3|
>
> We will add this result to OOD section in the revised version.
>
> >The comparison with baseline methods remains relatively limited...
>
> First, we would like to emphasize that the central goal of our work is to introduce a new foundational framework for model training. The proposed coded-smoothing module offers a key advantage: it applies to both **supervised and unsupervised** learning, whereas Mixup and its variants are inherently limited to supervised settings. This restriction arises because Mixup-style methods lack a **decoding (demixing) mechanism**. In contrast, coded smoothing explicitly penalizes the discrepancy between a model’s output with and without passing through the coded module. Without a decoding/remixing stage, Mixup and its extensions remain fundamentally restricted to supervised learning and operate only during the training phase—not during inference.
>
> Designing an effective mixing/demixing strategy—one that induces a meaningful discrepancy reflecting the smoothness of intermediate representations—is a nontrivial challenge. Our approach leverages principles from the broader field of coded computing to construct such a mechanism in a principled manner.
>
> Insted, our comparisons are structured along two fair and meaningful dimensions:
>
> **(1) Scope of Application:**
> Unlike Mixup, the coded-smoothing framework applies to both supervised and unsupervised tasks and can be used consistently during both training and inference.
>
> **(2) Performance:**
> Relative to Mixup/Manifold Mixup, our method achieves better or comparable generalization performance and superior robustness, while providing a broader and more principled theoretical foundation.
>
> We also remind the reviewers that Mixup was introduced in 2017 and has since inspired many variants, making exhaustive comparison impractical. However, in addition to Mixup, we include **Manifold Mixup** as another benchmark. Using the CIFAR-10 dataset and PreActResNet18 with the same hyperparameters as in Table 1 of the paper, and following the Manifold Mixup paper’s recommendation of $\alpha = 0.2$ for CIFAR10, we obtain the following results:
>
>
> | Method| CIFAR-10 (PARN18) |
> |-|-|
> | ERM | 93.8 |
> | Mixup | 95.6 |
> | Manifold Mixup | 95.4 |
> | **Coded-Smoothing** |95.8 |
>
> These results show that coded smoothing continues to outperform these alternatives. We will add manifold mixup as another benchmark to table 1 and 3 to the revised version of the paper.
>
>
> >The experimental evaluation of the generative model is currently insufficient...
>
> As the reviewer suggested, we will include generated outputs from a WGAN trained with the coded-smoothing module alongside those from a standard WGAN for qualitative comparison in the revised manuscript.
>
> Regarding the comparison with Mixup, we would like to emphasize that Mixup and other mixing-based methods cannot be applied to the generator, as they rely on label information. In the literature, Mixup and its variants are applied exclusively to the discriminator. In contrast, the novel decoding mechanism introduced in the coded-smoothing module enables us to regularize the generator as well. As shown in Table 2, this additional regularization improves the generalization performance of WGANs across multiple datasets. Thus, this comparison is infeasible and in fact shows the strenght of the proposed appraoch.

---

> > ### Author Response · Authors · 2025-11-21
> >
> > >The data encoding and decoding process introduces dimensionality expansion to the input...
> >
> > We thank the reviewer for this observation. The computational complexity of the coded-smoothing module is discussed in Appendix C of the manuscript.
> >
> > The coded path has approximately the same computational cost as the original path. The only extra overhead comes from the encoding and decoding operations (i.e., evaluating and fitting the splines), and these contribute only a negligible cost compared to the main  computation.
> >
> > More specifically, if the input and output dimensions are $d$ and $m$ (for example, when applying coded smoothing to the entire network on CIFAR-10, the input dimension is $32 \times 32$ and the output dimension is $10$), the computational complexities of the encoding and decoding steps are $\mathcal{O}((N+K)\cdot d)$ and $\mathcal{O}((N+K)\cdot m)$, respectively. Assuming $N \approx K$, both terms scale linearly in the dimension and batch size and are negligible relative to the main computation.
> >
> > To illustrate this in practice, we will include a **runtime analysis** in the revised manuscript. The s time for processing a batch of 128 CIFAR-10 images using PreActResNet18 on a single NVIDIA RTX 5090 GPU is:
> >
> > | Method         | Time                 |
> > |-|-|
> > | Original Path  | $1.5 \text{ ms} \pm 0.5$ |
> > | Coded Path     | $2 \text{ ms} \pm 0.4$   |
> >
> > These results show that the coded-smoothing module introduces only minimal computational overhead in practice.
> >
> > We will add more details on computational complexity including explanation and above runtime analysis.

---

### Official Review · Reviewer_L9et · 2025-10-28

**Soundness:** 3
**Presentation:** 3
**Contribution:** 2
**Rating:** 2
**Confidence:** 4

**Summary:**

This paper proposes a coded-smoothing module to enhance generalization in image classification.
Inspired by coded computing (Moradi et al., 2024), the authors design a regularization mechanism that promotes smoother feature representations.
Specifically, the module uses spline-based encoder and decoder functions: (1) the spline encoder is fitted to a batch of inputs x; (2) the encoded batch is processed by the main network f; and (3) the decoded outputs are compared with the originals to enforce consistency.
This process imposes a higher-order smoothness constraint on f, leading to better generalization.
Experiments are conducted on supervised image classification under in-domain, adversarial, and covariate-shift settings, as well as an unsupervised image-generation task.

**Strengths:**

- The paper introduces a novel perspective by adapting coded computing ideas, originally from distributed and information theory, into the context of ML regularization.

- The proposed coded-smoothing module is simple, lightweight, and applicable to both supervised and unsupervised settings.

- The spline-based encoder/decoder enforces higher-order smoothness, providing an interesting theoretical link between coding theory and representation regularization.

**Weaknesses:**

- Adversarial robustness results are weaker than MixUp. Under covariate shift, performance is also not better than Mixup.

- Missing comparisons and discussions with Manifold Mixup and related methods that already smooth latent representations. Table 1 and 3 should include full results of Manifold Mixup.

- Conceptual novelty is limited: the idea closely resembles Latent/Manifold Mixup and other interpolation-based regularizers.

- The inference method uses batch-based encoding/decoding (rather than single-sample). This reduces practicality in scenarios where only one input comes at a time.

- Notation and clarity issues: several functions (e.g., g1, g2) are undefined, making parts of the paper difficult to follow.

**Questions:**

see weaknesses

---

> ### Author Response · Authors · 2025-11-20
>
> We sincerely thank the reviewer for their thoughtful and encouraging feedback.
>
> **Weaknesses:**
> > Adversarial robustness results are weaker....
>
> **Answer:** We respectfully note that this characterization does not fully align with the results presented in the paper.
>
> Regarding adversarial robustness, Table 3 shows that our RCI approach (rows 2 and 3) significantly outperforms standard methods such as ERM, Mixup, and Manifold Mixup (row 1).
>
> Using RCI with the model trained with coded smoothing module is the best choice for all attacks except PGD 100 step (in this case RCI + coded smoothing is the second best). Under PGD-100, all of standard baselines drop to 0% accuracy, while RCI reaches **37%** accuracy when trained with Mixup and **27%** when trained with coded smoothing.
>
> Overall, these results demonstrate that RCI delivers substantial robustness improvements without using any adversarial training.
>
>
> For out-of-distribution evaluation (Table 4), coded smoothing outperforms in all datasets, except CIFAR-10.C, where Mixup shows a slight advantage. We further extend the OOD evaluation to include ImageNet-R dataset for ResNet-50 model trained on ImageNet-1K using different approaches. The results (to be added in the revised manuscript) are:
>
> | Method           | ImageNet-R Acc |
> |------------------|----------------|
> | Coded Smoothing  | **22.8**       |
> | Mixup            | 19.8           |
> | Raw              | 21.3           |
>
> Taken together, these results show that coded smoothing is stronger in most tasks under distribution-shift conditions and RCI method consistently outperforms other baselines in all types of attacks.
>
> In terms of application scope, coded smoothing also offers a broader range of use cases. Specifically, it can be applied to **unsupervised learning**, whereas Mixup cannot.
>
>
> > Missing comparisons and discussions with Manifold Mixup ...
>
> **Answer:** As requested, we implement the manifold mixup and add to our benchmark. These results will included in the revised manuscript and will be added to both Table 1 and Table 3. For PreActResNet18 on Cifar10 dataset the experiments is as follows:
>
> | Method           | CIFAR-10 (PARN18) |
> |------------------|--------------------|
> | ERM              | 93.8         |
> | Mixup            | 95.6         |
> | Manifold Mixup   | 95.4         |
> | **Coded-Smoothing** | 95.8  |
>
> As above tables shows, the proposed scheme outperforms manifold mixup too.
>
> > Conceptual novelty is limited...
>
>
> **Answer:** Let us clarify the novelty of our approach relative to MixUp and its variants:
>
> (1) MixUp performs only a mixing operation, whereas coded smoothing introduces both a mixing and a demixing (decoding) process. After demixing, the model’s output closely matches the output of the original model without any mixing/demixing layers.
>
> (2) The quality of this approximation can be considered as a proxy of the smoothness of the model. We incorporate the discrepancy between the two outputs into the loss function, thereby explicitly promoting smoothness during training.
>
> (3) Designing a mixing–demixing pair with minimal computational overhead and these smoothness properties draws on principles from general coded computing—a perspective entirely absent in MixUp-based methods.
> These properties enable coded smoothing to be used in both supervised and unsupervised settings, and in both training and inference. In contrast, MixUp and its variants are fundamentally restricted to supervised learning and only the training phase.
>
> In summary, coded smoothing module offers the following benefits.
>
> 1. It provides **consistent generalization improvements** over Mixup and Manifold Mixup across most experiments.
> 2. It does not need labels and therfere it is applicable to **unsupervised models**.
> 3. It can be applied to **any submodule** of a neural network, not only to inputs or feature layers.
> 4. It is supported by **theoretical guarantees** rooted in coded computing.
> 5. It enables **Randomized Coded Inference (RCI)**, which significantly outperforms the adversarial robustness without adversarial training or any computation/time overhead.
>
>
> > The inference method uses batch-based encoding/decoding (rather than single-sample)...
>
> **Answer**
> We acknowledge that the proposed RCI module can be used during batch inference; however, it is not required to use the coded path during inference. Even when coded smoothing is applied only during training, it still improves inference performance (See Table 1 and 2).
>
> Moreover, our method remains effective with small batch sizes. As shown in Table 7, it performs well with batch sizes as small as 8, which helps mitigate practicality concerns in many real-world settings.
>
> > Notation and clarity issues...
>
> **Answer:** Thank you for pointing out the notation and clarity issues in the manuscript. In the revised manuscript, we will clarify these notations and provided formal definitions for all functions that were previously undefined.

---

> > ### Comment · Reviewer_L9et · 2025-11-24
> > **Reply to Authors**
> >
> > Thank you for taking the time for your thorough response.
> >
> > ---
> >
> > ### [W1] Adversarial robustness results are weaker....
> >
> >
> > The reason I find the robustness results weak is that adversarial robustness evaluation is known to be unreliable unless strong evaluation protocols are used, such as AutoAttack or 100-step PGD with restarts (e.g., [Croce+2020], [Schlarmann+2024]). Evaluations on FGSM and small-step PGD (e.g., PGD-10) are no longer considered sufficient in recent robustness studies.
> > In Table 3, the proposed Coded-Smoothing underperforms Mixup under the stronger PGD-100 setting. Combined with the absence of AutoAttack or an ensemble of diverse attacks, I therefore do not regard the results in Table 3 as strong evidence of adversarial robustness.
> >
> >
> > [Croce+2020] Croce, Francesco, and Matthias Hein. "Reliable evaluation of adversarial robustness with an ensemble of diverse parameter-free attacks." International conference on machine learning. ICML 2020.
> >
> > [Schlarmann+2024] Schlarmann, Christian, et al. "Robust clip: Unsupervised adversarial fine-tuning of vision embeddings for robust large vision-language models." ICML 2024.
> >
> > > For out-of-distribution evaluation (Table 4), coded smoothing outperforms in all datasets, except CIFAR-10.C
> >
> > I do not find this claim convincingly supported. Considering the standard deviations reported in Table 4, the differences between Mixup and Coded-Smoothing are small and often within one standard deviation. This makes it difficult to argue for a clear and statistically meaningful improvement.
> >
> > Similarly, for the in-distribution results in Table 1, the improvements over Mixup appear modest and largely within the reported standard deviations. For this reason, I am not convinced that these experiments demonstrate a substantial generalization gain.
> >
> > ---
> >
> > ###  [W2] Missing baselines
> >
> > I am still not convinced by the limited newly added results.
> > Additionally, the result is not statistically significant.
> >
> > ---
> >
> > ###  [W3] Conceptual novelty
> >
> > Mixing and a demixing (decoding) process are indeed different from Mixup, and I appreciate this clarification.
> >
> > However, I still have reservations about the overall contribution:
> >
> > - While Coded-Smoothing is a new module, the empirical gains over standard baselines appear modest and are not clearly statistically significant, as noted above.
> > - Regarding Randomized Coded Inference (RCI), I find the incremental novelty limited. There already exist many randomization-based defenses that aim to improve robustness at inference time through stochasticity or gradient obfuscation (e.g., [Pinot+2019], [Taran+2019]). Moreover, [Athalye+2018] has shown that such defenses often rely on obfuscated gradients and require careful evaluation with BPDA or EOT-style attacks. The current paper does not discuss or compare against prior randomization-based defenses, nor does it provide an evaluation with EOT or other adaptive attacks. This raises the concern that the observed robustness gains from RCI may largely stem from gradient masking rather than genuinely more robust decision boundaries.
> >
> > [Pinot+2019] Pinot, Rafael, et al. "Theoretical evidence for adversarial robustness through randomization." Advances in neural information processing systems 32 (2019).
> >
> > [Taran+2019] Taran, Olga, et al. "Defending against adversarial attacks by randomized diversification." Proceedings of the IEEE/CVF Conference on Computer Vision and Pattern Recognition. 2019.
> >
> > [Athalye+2018] Athalye, Anish, Nicholas Carlini, and David Wagner. "Obfuscated gradients give a false sense of security: Circumventing defenses to adversarial examples." International conference on machine learning. PMLR, 2018.
> >
> > ---
> > Overall, I am not convinced by the significance of this work in its current form, mainly due to the limited evaluation, the lack of statistically strong gains over baselines, and the absence of comparisons and adaptive attacks for the randomization-based inference scheme. Please let me know if I have misunderstood any of the experimental settings or claims.

---

> ### Author Response · Authors · 2025-11-28
>
> We thank the reviewer for the complete and thorough feedback.
>
> ---
>
> ### Adversarial Robustness: The Gain Is Significant
>
> Regarding the adversarial robustness results, we would like to clarify the outcome:
>
> From Table 3, under the PGD-100 attack, **Mixup achieves only 0.4% accuracy**, while our proposed Randomized Coded Inference (RCI) method reaches **up to 37% accuracy**. This substantial improvement directly stems from the randomized encoding introduced by the coded smoothing module.
>
> In multiple sub-rows of Row 2 of Table 3, we further report the inference accuracy of RCI **when the model is trained with a different approach**. Even when the model is trained by coded smoothing, RCI still achieves **27% accuracy under PGD-100**, which **significantly outperforms Mixup (0.4%)**. We emphasize that Mixup achieves only **0.4%** under PGD-100, and **is significantly outperformed by any solution that uses our proposed RCI at inference time**.
>
> In Row 3, we present similar results to those in Row 2 when $N = 1.5K$ in RCI.
>
> ---
>
> ### Clarification of the Gains on Distribution Shift
>
> We acknowledge that improvements on CIFAR-10.1 and CIFAR-10-C are marginal or on par with Mixup. However, our method provides **statistically significant gains** on other benchmarks:
>
> - ImageNet-R: Mixup leads to a **1.5% drop** compared to ERM, whereas coded smoothing yields a **1.5% improvement**.
> - CIFAR-10.2: Mixup achieves **$85.7 \pm 0.2$**, while coded smoothing reaches **$86.4 \pm 0.1$**.
>
> This gain is considered **statistically significant and highly valued in the literature**.
>
> ---
>
> ### Novelty
>
> We thank the reviewer for acknowledging the novelty of the proposed method.
> Below, we elaborate on why RCI is fundamentally different from prior randomization-based approaches.
>
> Compared with other randomized inference methods such as Randomized Smoothing [1] and Randomized Diversification [2], which introduce randomness on each input and **require many forward passes per prediction**, RCI is significantly more efficient:
>
> - In [1] (page 34), under a **PGD-20** attack with radius 1 (same setting as ours), the best accuracy on CIFAR-10 is **22%**, obtained using 100 Monte Carlo samples — resulting in **100× inference cost**, which is prohibitive for large networks.
> - In contrast, **RCI incurs no additional inference cost when $N = K$**, and only **1.5× overhead when $N = 1.5K$** — dramatically lower than the **100× overhead** in randomized smoothing.
> - Moreover, **RCI achieves 37% accuracy under the much stronger PGD-100 attack**, compared to **22% under PGD-20** for randomized smoothing [1], demonstrating both **higher robustness and lower computational cost**.
>
> Similarly, [2] employs transformations such as DCT, random sign flipping, and inverse transforms, further increasing computational complexity to $I \cdot J \cdot d \cdot \log(d)$, where $J$ is the number of transformations per sample and $I$ is the number of channels. Note that results in [2] are **not comparable** to ours, as they do not evaluate their approach under strong attacks such as PGD-100.
>
> These results demonstrate a **fundamental distinction** between RCI and prior randomized inference approaches.
> We will add a discussion on this point in the revised version of the paper.

---

> > ### Author Response · Authors · 2025-11-28
> >
> > ### **On the Empirical Gains**
> >
> > We would like to emphasize the following key points:
> >
> > 1. **Marginal improvements in generalization error are common—and still highly valued—in this domain.**
> >    In the literature, gains on benchmark datasets such as CIFAR-10/100 and TinyImageNet are often marginal. For instance, Table 1 in RegMixup (NeurIPS 2022) [3] reports only a 0.4% accuracy improvement, without statistical indication. Likewise, Table 2 in SK Mixup (ICLR 2025) [4] shows a 0.4% accuracy gain over Mixup. Still, these papers offer other valuable contributions—for example, RegMixup improves performance under OOD, and SK Mixup is beneficial for calibration.
> >
> > 2. **Our proposed method offers significant gains in generalization error and distribution shift performance for some datasets, and minor gains for others. However, the gain in adversarial robustness is highly significant and valuable.**
> >    While coded smoothing yields marginal improvements on some in-distribution and out-of-distribution benchmarks (e.g., CIFAR-10/100 and CIFAR10-C), it demonstrates **notable** improvements on others, including **TinyImageNet, CIFAR-10.1, and CIFAR-10.2**.
> >
> > Additionally, coded smoothing offers the following highly significant improvements:
> >
> > - **(I) Significant adversarial robustness with *minimal* additional computational overhead.**
> >   For example, Mixup achieves **39.9%** accuracy under PGD-10, whereas RCI achieves **72%**. Under PGD-100, Manifold Mixup and Mixup achieve **0%** and **0.4%**, respectively — while RCI reaches **37%**, demonstrating a substantial robustness advantage. All of this occurs with **no additional computational cost when $N = K$**.
> >   When **$N = 1.5K$**, the inference complexity increases only by a factor of **1.5**.
> >
> > - **(II) Broader applicability to domains where existing regularization methods cannot be used**, such as regularizing generator networks in GANs.
> >
> > ---
> >
> > ### **Final Remark**
> >
> > Given the acknowledged **novelty**, **strong adversarial robustness**, and **practicality of the method**, we believe that maintaining a **strong reject (2/10)** score does not fully reflect the merits of the contribution.
> >
> > ---
> >
> > **References**
> >
> > [1] Cohen, J., et al. *Certified adversarial robustness via randomized smoothing.* ICML 2019.
> > [2] Taran, O., et al. *Defending against adversarial attacks by randomized diversification.* ICCV 2019.
> > [3] Pinto, F., et al. *Using mixup as a regularizer can surprisingly improve accuracy & out-of-distribution robustness.* NeurIPS 2022.
> > [4] Bouniot, Q., et al. *Tailoring Mixup to Data for Calibration.* ICLR 2025.

---

### Official Review · Reviewer_caW3 · 2025-11-01

**Soundness:** 3
**Presentation:** 3
**Contribution:** 3
**Rating:** 6
**Confidence:** 3

**Summary:**

This paper introduces **Coded-Smoothing**, a new regularization module inspired by **coded computing theory**, which traditionally mitigates straggler and adversarial failures in distributed computing by operating on coded (linearly combined) data. The authors adapt this principle to deep learning, proposing a module that encourages **local smoothness** and **better generalization**, while also improving **adversarial robustness**.

The proposed module operates in three steps:
1. **Encoding** — combines a batch of inputs into coded samples via spline interpolation.
2. **Computation** — evaluates the model on the coded samples.
3. **Decoding** — reconstructs approximate outputs of the original samples and penalizes discrepancies to enforce smoothness.

During inference, the authors propose **Randomized Coded Inference (RCI)**, which randomizes the input order before encoding to disrupt gradient-based adversarial attacks (e.g., FGSM, PGD).

Experiments on CIFAR-10/100, TinyImageNet, and GANs demonstrate that coded-smoothing improves generalization compared to ERM and Mixup, enhances adversarial robustness, and applies effectively in both supervised and unsupervised settings, all with minimal computational overhead.

**Strengths:**

- **Novel conceptual link:** The paper establishes a creative and original connection between **coding theory** and **regularization in deep learning**, introducing a theoretically motivated way to enforce smoothness.

- **Unified applicability:** The proposed coded-smoothing module works seamlessly for **both supervised and unsupervised** learning settings, unlike Mixup-style approaches that rely on label information.

- **Strong empirical results:** Demonstrates consistent improvements in test accuracy and adversarial robustness across multiple datasets (CIFAR-10/100, TinyImageNet) and architectures.

- **Minimal computational overhead:** The spline-based implementation adds negligible cost to training and inference, making the method practical for large-scale applications.

- **Adversarial robustness:** The **Randomized Coded Inference (RCI)** strategy offers a simple yet effective defense against gradient-based adversarial attacks without requiring adversarial training.

- **Theoretical grounding:** Provides analytical justification (Lemma 1) that links the number of coded samples and function smoothness to the approximation error, offering theoretical intuition for why the method works.

**Weaknesses:**

- **Narrow comparative evaluation:** Experiments mainly compare against ERM and Mixup. Other strong baselines such as CutMix, Manifold Mixup, consistency regularization, or adversarial training are missing.

- **Hyperparameter interpretability:** The effects of key hyperparameters (e.g., the weighting factor µ and the ratio of coded samples N/K) are not clearly analyzed or justified, and practical tuning guidance is lacking.

- **Ablation study depth:** Although appendices mention ablation analyses, the main paper does not clearly quantify trade-offs or sensitivity regarding batch size, coded sample count, or the balance between smoothness and accuracy.

- **Assumption of smoothness transfer:** It is assumed that spline-induced smoothness in the input domain translates to smoother representations in feature space, but this relationship is not empirically validated.

- **Robustness evaluation limitations:** The reported adversarial robustness results do not appear to test **adaptive attacks** (e.g., expectation-over-transformation), which may overestimate the protection offered by RCI.

**Questions:**

1. **Gradient flow and differentiability:**
   The paper provides a pseudo-code implementation of the coded-smoothing module but does not clarify how gradients flow through the encoder and decoder (spline fitting) steps.
   Are these spline operations treated as differentiable with respect to the network parameters, or are they non-trainable transformations through which gradients do not propagate?
   Have you observed any gradient stability issues due to the decoding approximation?

2. **Fixed versus random coding points:**
   The encoding and decoding points (Chebyshev nodes) appear to be fixed throughout training.
   Have you experimented with randomizing or re-sampling these points during training, or toward the end of training, to act as an additional stochastic regularizer?
   Could dynamic or randomized coding points improve generalization or robustness?

3. **Theoretical connection:**
   Lemma 1 provides intuition about the approximation error, but it remains unclear how the proposed regularization quantitatively affects model smoothness or generalization.
   Can you provide a more formal connection between coded-smoothing and a Lipschitz or higher-order smoothness bound on the function \( f(\cdot) \)?

4. **Adversarial robustness evaluation:**
   The reported results demonstrate strong performance under FGSM and PGD attacks.
   Have you tested the method against **adaptive attacks** that account for inference-time randomness (e.g., expectation-over-transformation)?
   How does the robustness change when such attacks are considered?

5. **Hyperparameter sensitivity:**
   How sensitive is performance to the choice of key parameters such as the weighting factor \( \mu \), the ratio \( N/K \), and batch size?
   Can you offer empirical or theoretical guidance on tuning these parameters?

6. **Selective application:**
   The paper applies coded-smoothing to the full network.
   Have you explored applying it only to specific layers or blocks to trade off computational cost and regularization strength?

7. **Comparison to other smoothness-based regularizers:**
   How does coded-smoothing relate to other smoothness-enforcing techniques like Jacobian regularization, spectral normalization, or consistency regularization?
   Could it be complementary to these methods?

---

> ### Author Response · Authors · 2025-11-21
>
> We sincerely thank the reviewer for their thoughtful and encouraging feedback.
> > 1. Narrow comparative evaluation....
>
> **Answer:**
> We would like to clarify that the primary objective of our work is to introduce a new foundational framework for model training that can be seamlessly integrated into both supervised and unsupervised pipelines. Our coded-smoothing module offers a key advantage over Mixup and its variants: unlike those methods, which only perform data mixing without any corresponding demixing or decoding stage, our approach incorporates an explicit decoding mechanism. This decoding process enables the model to approximately recover the original output, and the accuracy of this reconstruction serves as a meaningful measure of the model’s smoothness. The resulting discrepancy can therefore be used as a loss term to promote smoothness and improve generalization, allowing coded smoothing to apply to a much broader range of tasks. In contrast, Mixup-style methods remain inherently restricted to supervised learning precisely because they lack a decoding or “unmixing” step.
>
> Designing effective mixing/demixing (or coding/decoding) stages that (1) remain computationally efficient and (2) induce discrepancies that meaningfully capture the smoothness of the underlying model is a delicate task—an “art” informed by principles from generalized coded computing. This perspective is entirely absent from Mixup and its variants. Moreover, Mixup, introduced in 2017 and now with over 14,000 citations, has spawned many variants, making exhaustive comparison infeasible.
>
> To extend our comparison, we include **Manifold Mixup** as another strong benchmark. Using the CIFAR-10 dataset and PreActResNet18 with $\alpha = 0.2$ (as recommended in the Manifold Mixup paper), we obtain:
>
> | Method           | CIFAR-10 (PARN18) |
> |------------------|--------------------|
> | ERM              | 93.8         |
> | Mixup            | 95.6         |
> | Manifold Mixup   | 95.4        |
> | **Coded-Smoothing** |95.8      |
>
> These results show that the proposed method continues to outperform mixing-based approaches. We will add this result to the revised manuscript.
>
> Regarding adversarial robustness, our **Randomized Coded Inference (RCI)** procedure—built upon the coded-smoothing module—significantly improves robustness *without* requiring adversarial training. As noted in the Mixup and Manifold Mixup papers and as shown in Table 3, these methods perform poorly under FGSM and especially PGD attacks (e.g., under 100-step PGD, their accuracy drops to nearly zero). In contrast, our RCI procedure (rows 2 and 3 in Table 3) substantially enhances adversarial robustness while remaining training-free.
>
> Investigating other classes of attacks is an interesting direction that we plan to pursue in future work.
>
> We hope this clarifies the distinctions and advantages of our approach. We will update the manuscript to reflect these comparisons and improvements.
>
>
> > 2. Hyperparameter interpretability....
>
> **Answer:** Thank you for your question regarding the hyperparameters. We have discussed hyperparameter selection in detail in Appendix D.1.1 of the manuscript, where we also provide our recommended default values. These values were empirically selected to perform well across multiple datasets and architectures.
>
> To summarize, we empirically find that choosing $N = K = \text{batch size}$ and $\mu = 0.5$ provides the best trade-off between regularization strength and predictive accuracy. Additionally, we recommend a scheduling strategy for $N$, where $N$ is gradually increased to $1.5K$ during the final training epochs (see Section D.1.1 and Figure 5(a) in the appendix).
>
> Furthermore, Appendix G presents a detailed sensitivity analysis of $N$ and $\mu$, examining their impact on model performance (Table 5,6,7 in Section G of Appendix) . This analysis highlights how these hyperparameters interact and provides guidance on how to fine-tune them for different settings.
>
> > 3. Ablation study depth...
>
> **Answer:** Appendix G presents a detailed sensitivity analysis of $N$ and $\mu$, examining their impact on model performance. Specifically, Table 5 analyzes the effect of the number of coded points $N$, Table 6 investigates the effect of $\mu$ and the trade-off between inducing smoothness and maintaining accuracy, and Table 7 reports the impact of the inference-time batch size on performance.

---

> > ### Author Response · Authors · 2025-11-21
> >
> > > 4. Assumption of smoothness transfer...
> >
> > We would first like to clarify that, while smoothness is an important aspect of our approach, it is not the final objective. The primary goal of the coded-smoothing module is to improve generalization and enhance robustness—not merely to enforce smoothness. The smoother representations produced by our method serve this broader purpose, ultimately leading to improved generalization across different tasks.
> >
> > We support this claim through both empirical and theoretical evidence. Empirically, Figure 1(b) shows that applying the coded-smoothing module results in noticeably smoother decision boundaries in the feature space. Theoretically, Lemma 1 establishes that when the decoded representations remain close to the model’s original outputs, the function becomes smoother. Together, these results demonstrate that training with the coded-smoothing module consistently induces smoother behavior, which in turn contributes to improved generalization and robustness.
> >
> > > 5. Robustness evaluation limitations...
> >
> > We acknowledge that our current experiments focus on gradient-based adversarial attacks (e.g., FGSM and PGD). It is important to note that existing mixing-based methods, including Manifold Mixup, experience a significant performance drop under such attacks—both Mixup and Manifold Mixup degrade to nearly zero accuracy under PGD-10. Thus, achieving robustness for these baselines remains a challenging open problem.
> >
> > Evaluating robustness against more advanced adaptive attacks is a natural next step for further validating the effectiveness of our randomized coded inference (RCI) method. We plan to extend our robustness evaluation in future work by incorporating such adaptive attacks, thereby providing a more comprehensive assessment of the protection offered by RCI.
> >
> > **Questions:**
> > > 1. Gradient flow and differentiability....
> >
> >  In coded-smoothing module, the encoder and decoder are based on smoothing splines, which are inherently differentiable. The encoder produces coded samples by applying a weighted linear combination of the input samples, and the decoder reconstructs the estimated outputs based on these coded samples. Specifically, both encoding and decoding operations can be represented in matrix form, allowing us to treat these operations as differentiable with respect to the network parameters.
> > To provide more detail, in the revised manuscript, we will clarify this as follows:
> >
> > Following equation (6), we define
> >
> > $A^{\text{enc}} := A_{\vec{\alpha},\vec{\beta}} \in \mathbb{R}^{K \times N}$ and $A^{\text{dec}} := A_{\vec{\beta},\vec{\alpha}} \in \mathbb{R}^{N \times K}$. Thus $\forall j \in [N], \quad \tilde{x}_j =  \langle A_j^{\text{enc}}, \vec{x} \rangle$
> > where $A_j^{\text{enc}}$ denotes the $j$th column of the matrix $A^{\text{enc}}$ and $\vec{x}=[x_1, \ldots, x_K]^T$.  Finally, letting $\vec{f}(A^{\text{enc}}, \vec{x}) := [f(\langle A_1^{\text{enc}},\vec{x} \rangle),\dots,f(\langle A_N^{\text{enc}},\vec{x} \rangle)]^T$, the decoded estimates are given by $\forall i \in [K], \quad \hat{f}(x_i) = \langle A_i^{\text{dec}}, \vec{f}(A^{\text{enc}}, \vec{x}) \rangle$
> >
> > Regarding gradient stability, we have not observed any issues during training with the decoding approximation. Our experiments have demonstrated stable gradient flow and efficient optimization, indicating that the spline-based encoding and decoding process does not interfere with the overall training procedure.
> >
> > > 2. Fixed versus random coding points...
> >
> > We thank the reviewer for the interesting suggestion. In practice, the stability of interpolation is highly dependent on the minimum consecutive distance between the encoding or decoding points. Therefore, selecting these points fully at random would make the module numerically unstable. However, randomly sampling from a set of well-designed, stable points can serve as an additional source of randomness. We have experimented with this approach but did not observe any notable improvement in performance.
> >
> > > 3. Theoretical connection...
> >
> > While Lemma 1 provides an intuitive understanding of the approximation error, we acknowledge that a more formal connection to model smoothness or generalization could further clarify the impact of our method. However, we would like to emphasize that the primary goal of the coded-smoothing module is not to directly enforce smoothness, but rather to improve generalization. A formal connection to Lipschitz or higher-order smoothness bounds would be an interesting direction for future work, but it is outside the scope of this current paper.
> >
> > > 4. Adversarial robustness evaluation ...
> >
> >  As mentioned in our earlier response, we have not yet tested our method against adaptive attacks that account for inference-time randomness, such as expectation-over-transformation. However, we plan to extend our robustness evaluation in future work by including such attacks to assess how they affect the robustness of our method.

---

> > > ### Author Response · Authors · 2025-11-21
> > >
> > > > 5. Hyperparameter sensitivity...
> > >
> > > **Answer:** Please refer to the answer of weakness 2.
> > >
> > > > 6. Selective application....
> > >
> > > **Answer:** We have already explored this idea in the context of GANs, where the coded-smoothing module was applied selectively to the generator. Based on this, we also conducted additional experiments, including an ablation study, where we selectively applied coded-smoothing to different layers or blocks within the ResNet network. We apply coded smoothing module to different set of blocks in the PreActResNet18 for the cifar10 tasks. The result is as follows and will be in the revised manuscript:
> > >
> > >
> > > | Set of blocks in the coded path| Test Acc | Test Loss |
> > > |-|-|-|
> > > | 0                    | 94.6  |  0.25  |
> > > | 3                    |  94.6 |  0.26  |
> > > | 0,1                  | 94.8  |  0.23  |
> > > | 0,2                  | 94.8  |  0.23  |
> > > | 1,3                  |  94.8 |  0.235  |
> > > | 0,1,2                |  94.5 | 0.24   |
> > > | 0,1,2,3              |  95.1 |   0.19 |
> > > | 0,1,2,3, 4              |  95.9 |   0.19 |
> > >
> > > > 7. Comparison to other smoothness-based regularizers:
> > >
> > > **Answer:** Analyzing the theoretical connections between our method and other smoothness-inducing regularizers is an interesting future direction that we plan to explore.

---

> > > > ### Comment · Reviewer_caW3 · 2025-11-26
> > > >
> > > > I appreciate the thorough explanations provided regarding the questions raised. However, I will maintain my original score. My primary reason is that the performance improvements demonstrated by the module do not appear sufficiently significant to warrant a change in assessment.

---

> > > > > ### Author Response · Authors · 2025-11-28
> > > > >
> > > > > We thank the reviewer for **acknowledging the novelty and portability** of the proposed coded smoothing module. As a final note, we would like to emphasize the following points for clarity:
> > > > >
> > > > > 1. **Marginal improvements in generalization error are common—and still highly valued—in this domain.**
> > > > >    It is worth noting that, in the literature, gains on benchmark datasets such as CIFAR-10/100 and TinyImageNet are often marginal. For instance, Table 1 in RegMixup (NeurIPS 2022) [1] reports only a 0.4% accuracy improvement, without statistical indication. Likewise, Table 2 in SK Mixup (ICLR 2025) [2] shows a 0.4% accuracy gain over Mixup. Still, these papers offer other important contributions—for example, RegMixup improves performance under OOD, and SK Mixup is beneficial for calibration.
> > > > >
> > > > > 2. **Our proposed method offers significant gains in generalization error and distribution shift performance for some datasets, and minor gains for others. However, the gain in adversarial robustness is highly significant and valuable.**
> > > > >    While coded smoothing yields marginal improvements on some in-distribution and out-of-distribution benchmarks (e.g., CIFAR-10/100 and CIFAR10-C), it demonstrates **notable** improvements on others, including **TinyImageNet, CIFAR-10.1, and CIFAR-10.2**.
> > > > >
> > > > > Additionally, coded smoothing offers following highly significant improvements:
> > > > > - **(I) Significant adversarial robustness with *minimal* additional computational overhead.**
> > > > >   For example, Mixup achieves **39.9%** accuracy under PGD-10, whereas our method (RCI) attains **72%**. Under PGD-100, Manifold Mixup and Mixup achieve **0%** and **0.4%**, respectively—while RCI reaches **37%**, demonstrating a substantial robustness advantage. All of this occurs with no additional computational cost when $N = K$.
> > > > >   When $N = 1.5K$, the inference complexity increases only by a factor of 1.5.
> > > > >
> > > > > - **(II) Broader applicability to domains where existing regularization methods cannot be used**, such as regularizing generator networks in GANs.
> > > > >
> > > > > In summary, although the gains on standard benchmarks are consistent with prior work in the field, the **unique strengths** of coded smoothing—particularly in adversarial robustness and applicability to otherwise unsupported architectures—reflect a meaningful and impactful contribution.
> > > > >
> > > > > ---
> > > > >
> > > > > **References**
> > > > > [1] Pinto, Francesco, et al. *“Using mixup as a regularizer can surprisingly improve accuracy & out-of-distribution robustness.”* NeurIPS 2022.
> > > > > [2] Bouniot, Q., et al. *“Tailoring Mixup to Data for Calibration.”* ICLR 2025.

---

### Official Review · Reviewer_TRqB · 2025-11-01

**Soundness:** 2
**Presentation:** 3
**Contribution:** 2
**Rating:** 6
**Confidence:** 3

**Summary:**

The paper proposes a regularization method: coded-smoothing module to regularize deep networks by enforcing local smoothness: a batch is encoded into multiple coded samples via spline/Chebyshev points, passed through a chosen network block, and then decoded to reconstruct the original block outputs; a reconstruction loss is added to the task loss, and at inference a randomized coded inference (RCI) variant permutes batches to disrupt gradient-based attacks; experiments on CIFAR-10/100, TinyImageNet, WGAN-GP (CIFAR-10, CelebA), and distribution shift (CIFAR-10.1/10.2/10C) show modest in-distribution gains over ERM/mixup, improved GAN IS/FID, and notable robustness boosts under FGSM/PGD (though not competitive with strong adversarial training), supported by a lemma that bounds decoding error improving with function smoothness and larger code size.

**Strengths:**

* The idea of reframing coded computing as a data/function-space regularizer is novel and goes beyond pairwise linearity such as mixup and SMOTE.
* The presentation is clear, provides a clean algorithmic description, a formal error bound for the coded reconstruction, and ablations/sensitivity in appendices.
* Clarity: the training objective is simple (one mixing coefficient μ) and the module integrates at arbitrary layers without label dependence.
* Leads to generalization improvements and inference-time robustness via RCI without retraining or adversarial training.

**Weaknesses:**

* The evaluation is limited to vision at CIFAR/TinyImageNet scale, with no ImageNet-1k or transformer/non-vision results, so cross-domain applicability remains uncertain.
* The run time impact is not reported. The proposed module contains mixed component both within module, in loss function and in data space. Therefore it is important to understand the impact on module latency and complexity.
* Visual messages in figures are often not clear. For example figure 1 (a) misses the core idea of code computing that enforcing closeness between decoded estimates and true outputs; and in figure 1(b) the decision boundary in both panels appear very similar.

**Questions:**

* Scaling/generalization: how does the method perform on ImageNet-1k and on larger transformers (e.g., ViT/BERT) where sequence batching and attention may interact with coded batches?
* Hyperparameter guidance: the method introduces various new hyperparameters, what robust default choices of N/K, μ, and spline/Chebyshev order work across datasets, or any recommended tuning strategy?
* Compute overhead: what are the training/inference time and memory overheads versus ERM/mixup across N and batch sizes, and how does RCI affect latency?
* Baseline breadth: how does the method compare with CutMix, AugMix, consistency regularization, and post-hoc smoothing/ensembling; any negative interactions?
* Dropout comparison: when combined with (or compared to) dropout/DropBlock/stochastic depth, are the gains additive, redundant, or conflicting, and at which layers should each be applied?
* Figure clarity: can you quantify the two-spirals boundary difference (e.g., curvature/total variation metrics, margin maps) to substantiate the qualitative claim?

---

> ### Author Response · Authors · 2025-11-21
>
> We sincerely thank the reviewer for their thoughtful and encouraging feedback.
> > 1. The evaluation is limited to vision...
>
> **Answer:**  We thank the reviewer for the thoughtful feedback. We would like to clarify that the primary objective of this work is to introduce a **new foundational module for training and inference pipelines** that integrates seamlessly into existing **supervised and unsupervised** ML frameworks.
>
> Our approach is fundamentally different from existing mixing-based methods such as Mixup and Manifold Mixup. In contrast to these techniques—which only perform mixing—we introduce an explicit **decoding (demixing) step** after mixing. This design brings several advantages:
>
> 1. It provides **consistent generalization improvements** over Mixup and Manifold Mixup across most experiments.
> 2. It does not need labels and therfere it is applicable to **unsupervised models**.
> 3. It can be applied to **any submodule** of a neural network, not only to inputs or feature layers.
> 4. It is supported by **theoretical guarantees** rooted in coded computing.
> 5. It enables **Randomized Coded Inference (RCI)**, which significantly outperforms the adversarial robustness without adversarial training or any computation/time overhead.
>
> Although our proof-of-concept experiments focus on CIFAR-10/100 and TinyImageNet, the framework is **not restricted** to these datasets. It can be incorporated into broader supervised and unsupervised pipelines, including more complex architectures such as ViTs and Transformers.
>
> To further strengthen the empirical evaluation, we additionally trained **ResNet-50 on ImageNet-1K** following the Mixup paper’s recommended settings (α = 0.2; 90 epochs; initial LR = 0.2 with decays at epochs 30, 60, 80). The results are:
>
> |Method|Test Loss|Test Acc|
> |-|-|-|
> | Coded Smoothing |**1.20**|**70.1**|
> | Mixup|1.26|69.1|
> | Raw| 1.25|69.5|
>
> We will include these ImageNet-1K results in the revised version.
>
> We acknowledge that covering all potential applications is beyond the scope of a single paper. Our goal here is to establish the **core foundation** and demonstrate its utility across representative supervised and unsupervised settings. Extending this framework to larger-scale models (e.g., Transformers) and additional domains (e.g., NLP) is part of future work.
>
>
> > 2. The run time impact....
>
> **Answer:** We thank the reviewer for this observation. The computational complexity of the coded-smoothing module is discussed in Appendix C of the manuscript.
>
> The coded path has approximately the same computational cost as the original path. The only extra overhead comes from the encoding and decoding operations (i.e., evaluating and fitting the splines), and these contribute only a negligible cost compared to the main  computation.
>
> More specifically, if the input and output dimensions are $d$ and $m$ (for example, when applying coded smoothing to the entire network on CIFAR-10, the input dimension is $32 \times 32$ and the output dimension is $10$), the computational complexities of the encoding and decoding steps are $\mathcal{O}((N+K)\cdot d)$ and $\mathcal{O}((N+K)\cdot m)$, respectively. Assuming $N \approx K$, both terms scale linearly in the dimension and batch size and are negligible relative to the main computation.
>
>
>
> To illustrate this in practice, we will include a **runtime analysis** in the revised manuscript. The s time for processing a batch of 128 CIFAR-10 images using PreActResNet18 on a single NVIDIA RTX 5090 GPU is:
>
> | Method         | Time                 |
> |----------------|----------------------|
> | Original Path  | $1.5 \text{ ms} \pm 0.5$ |
> | Coded Path     | $2 \text{ ms} \pm 0.4$   |
>
> These results show that the coded-smoothing module introduces only minimal computational overhead in practice.
>
> We will add more details on computational complexity including explanaintion and above runtime analysis.
>
>
>
> > 3. Visual messages in figures ....
>
> **Answer:**  Thank you for your feedback regarding the clarity of the figures. In response to your comment, we will make the following revisions:
>
> **Figure 1(a):**  We will update this figure to more clearly illustrate the core idea of the coded-smoothing module, with an emphasis on how the decoded estimates remain close to the true outputs, resulting in smoother representations.
>
> **Figure 1(b):**  We will add additional details to make the distinction between methods more apparent. In particular, we now include a zoom-in view to highlight the smoother decision boundaries produced by our method compared to the baseline. We also expand the figure caption to more clearly describe these differences in boundary geometry.

---

> > ### Author Response · Authors · 2025-11-21
> >
> > **Questions**
> >
> > > 4. Scaling/generalization?
> >
> > Please refer to the answer of first weakness. We have added new set of experiments on the imagenet-1k dataset.
> >
> > > 5. Hyperparameter guidance?
> >
> > **Answer:**
> > Thank you for your question regarding the hyperparameters. We have discussed hyperparameter selection in detail in Appendix D.1.1 of the manuscript, where we also provide our recommended default values. These values were empirically selected to perform well across multiple datasets and architectures.
> >
> > To summarize, we empirically find that choosing $N = K = \text{batch size}$ and $\mu = 0.5$ provides the best trade-off between regularization strength and predictive accuracy. Additionally, we recommend a scheduling strategy for $N$, where $N$ is gradually increased to $1.5K$ during the final training epochs (see Section D.1.1 and Figure 5(a) in the appendix).
> >
> > We use second-order splines due to their simplicity, low computational overhead, and strong regularization effect. For the encoding and decoding points, we employ first- and second-order Chebyshev points because of their well-established numerical stability in interpolation [1].
> >
> > Furthermore, Appendix G presents a detailed sensitivity analysis of $N$ and $\mu$, examining their impact on model performance (Table 5,6,7 in Section G of Appendix). This analysis highlights how these hyperparameters interact and provides guidance on how to fine-tune them for different settings.
> >
> > > 6. Compute overhead?
> >
> > **Answer:**
> > Please refer to our response to Weakness 2 for a detailed explanation of the inference and training time. Since the coded-smoothing module is non-parametric, it does not introduce any memory overhead. Additionally, as discussed in Section 5, RCI relies only on simple shuffling operations after encoding and before decoding, which incur no measurable latency during prediction.
> > We will add more discussion about all these computaion/time overhead to the revised paper.
> >
> > > 7. Baseline breadth?
> >
> > **Answer:**
> > As discussed earlier, the central goal of our work is to introduce a new foundational framework for model training. The proposed coded-smoothing module offers a key advantage: it applies to both **supervised and unsupervised** learning, whereas Mixup and its variants are inherently limited to supervised settings. This restriction arises because Mixup-style methods lack a **decoding (demixing) mechanism**. In contrast, coded smoothing explicitly penalizes the discrepancy between a model’s output with and without passing through the coded module. Without a decoding/remixing stage, Mixup and its extensions remain fundamentally restricted to supervised learning and operate only during the training phase—not during inference.
> >
> > Designing an effective mixing/demixing strategy—one that induces a meaningful discrepancy reflecting the smoothness of intermediate representations—is a nontrivial challenge. Our approach leverages principles from the broader field of coded computing to construct such a mechanism in a principled manner.
> >
> > We also remind the reviewers that Mixup was introduced in 2017 and has since inspired many variants. Exhaustively comparing with every derivative is not feasible. However, in addition to Mixup, we have added **Manifold Mixup** with $\alpha = 0.2$ (as suggested in the Manifold Mixup paper) as another benchmark. Using the CIFAR-10 dataset and PreActResNet18 with the same hyperparameters as in Table 1 of the main paper, we obtain:
> >
> > | Method| CIFAR-10 (PARN18) |
> > |-|-|
> > | ERM | 93.8 |
> > | Mixup | 95.6 |
> > | Manifold Mixup | 95.4 |
> > | **Coded-Smoothing** |95.8 |
> >
> > These results show that coded smoothing continues to outperform these alternatives.
> >
> > Our comparisons are structured along two fair and meaningful dimensions:
> >
> > **(1) Scope of Application:**
> > Unlike Mixup, the coded-smoothing framework applies to both supervised and unsupervised tasks and can be used consistently during both training and inference.
> >
> > **(2) Performance:**
> > Relative to Mixup, our method achieves better or comparable generalization performance and superior robustness, while providing a broader and more principled theoretical foundation.
> >
> >
> >  > 8. Dropout comparison ...?
> >
> > **Answer:** We use the coded-smoothing module both with and without dropout. Following the recommendation in [1], we add dropout after the second batch-normalization layer in each ResNet block. The results below show that dropout integrates seamlessly with the proposed module and provides a slight improvement in generalization performance.
> >
> > | Method(Cifar10, PreActResNet18)| Test Acc | Test Loss |
> > |-|-|-|
> > | Raw                    | 93.780   | 0.308     |
> > | Raw + Drop Out         | 93.840   | 0.325     |
> > | Coded Smoothing              | 95.120   | 0.240     |
> > | Coded Smoothing + Drop Out   | 95.200   | 0.217     |
> >
> >
> >  >9. Figure clarity...?
> >
> > **Answer:** Please refer to answer of weakness 3.
> >
> > [1] Kim, Bum Jun, et al. "How to use dropout correctly on residual networks with batch normalization." PMLR, 2023.

---

> > > ### Comment · Reviewer_TRqB · 2025-11-26
> > >
> > > I thank the authors for their effort in responding to all my initial concerns.  With the extended evidence, I believe the paper is:
> > >
> > > 1, conceptually novel and portable to various network domains.
> > > 2, has consistent but limited empirical benefit . While the author rebuttal has included larger benchmarks such as Imagenet-1k, the gap between code smoothing and baseline is small for both test loss (0.05) and test acc (1%), a magnitude consistent with the main paper.
> > >
> > > Therefore, I tend to maintain my score.

---

> > > > ### Author Response · Authors · 2025-11-28
> > > >
> > > > We thank the reviewer for **acknowledging the novelty and portability** of the proposed coded smoothing module. As a final note, we would like to emphasize the following points for clarity:
> > > >
> > > > 1. **Marginal improvements in generalization error are common—and still highly valued—in this domain.**
> > > >    It is worth noting that, in the literature, gains on benchmark datasets such as CIFAR-10/100 and TinyImageNet are often marginal. For instance, Table 1 in RegMixup (NeurIPS 2022) [1] reports only a 0.4% accuracy improvement, without statistical indication. Likewise, Table 2 in SK Mixup (ICLR 2025) [2] shows a 0.4% accuracy gain over Mixup. Still, these papers offer other important contributions—for example, RegMixup improves performance under OOD, and SK Mixup is beneficial for calibration.
> > > >
> > > > 2. **Our proposed method offers significant gains in generalization error and distribution shift performance for some datasets, and minor gains for others. However, the gain in adversarial robustness is highly significant and valuable.**
> > > >    While coded smoothing yields marginal improvements on some in-distribution and out-of-distribution benchmarks (e.g., CIFAR-10/100 and CIFAR10-C), it demonstrates **notable** improvements on others, including **TinyImageNet, CIFAR-10.1, and CIFAR-10.2**.
> > > >
> > > > Additionally, coded smoothing offers following highly significant improvements:
> > > > - **(I) Significant adversarial robustness with *minimal* additional computational overhead.**
> > > >   For example, Mixup achieves **39.9%** accuracy under PGD-10, whereas our method (RCI) attains **72%**. Under PGD-100, Manifold Mixup and Mixup achieve **0%** and **0.4%**, respectively—while RCI reaches **37%**, demonstrating a substantial robustness advantage. All of this occurs with no additional computational cost when $N = K$.
> > > >   When $N = 1.5K$, the inference complexity increases only by a factor of 1.5.
> > > >
> > > > - **(II) Broader applicability to domains where existing regularization methods cannot be used**, such as regularizing generator networks in GANs.
> > > >
> > > > In summary, although the gains on standard benchmarks are consistent with prior work in the field, the **unique strengths** of coded smoothing—particularly in adversarial robustness and applicability to otherwise unsupported architectures—reflect a meaningful and impactful contribution.
> > > >
> > > > ---
> > > >
> > > > **References**
> > > > [1] Pinto, Francesco, et al. *“Using mixup as a regularizer can surprisingly improve accuracy & out-of-distribution robustness.”* NeurIPS 2022.
> > > > [2] Bouniot, Q., et al. *“Tailoring Mixup to Data for Calibration.”* ICLR 2025.

---

### Meta-Review · Area_Chair_U8qj · 2026-01-06

**Summary:**

In this submission, the authors proposed to tackle the problem of generalization through coding theory, with a coded-smoothing module a mixup fashion. Reviewers raised questions regarding the novelty, as well as evaluations (which is the major concern raised by all reviewers).

**Reviewer Concerns:**

The AC read through the manuscript, reviewers' comments and rebuttal. To the AC, the concerns regarding the novelty and evaluations have not been fully addressed.

Novelty: the authors rebottled in a way focusing on experiments and improvement. However, it fails to explain why such coded-smoothing module benefit generalization theoritically, which is the main focus of the paper.

Evaluations: as pointed out by reviewers, the improvement is marginal, and only mixup-related baselines are considered as baselines. Please note in practice, one can have different types of OOD samples. Only evaluating on adversarial attack is far from satisfactory.

**Reviewer Scores:**

Reviewers are less likely to update the scores.

---

### Decision · Program_Chairs · 2026-01-26

Reject